# Proteomic characterization of the *Toxoplasma gondii* cytokinesis machinery portrays an expanded hierarchy of its assembly and function

Klemens Engelberg [1] ✉, Tyler Bechtel[2], Cynthia Michaud[1], Eranthie Weerapana [2] & Marc-Jan Gubbels [1] ✉

The basal complex (BC) is essential for *T. gondii* cell division but mechanistic details are lacking. Here we report a reciprocal proximity based biotinylation approach to map the BC's proteome. We interrogate the resulting map for spatiotemporal dynamics and function by disrupting the expression of components. This highlights four architecturally distinct BC subcomplexes, the compositions of which change dynamically in correlation with changes in BC function. We identify BCC0 as a protein undergirding BC formation in five foci that precede the same symmetry seen in the apical annuli and IMC sutures. Notably, daughter budding from BCC0 progresses bidirectionally: the apical cap in apical and the rest of the IMC in basal direction. Furthermore, the essential role of the BC in cell division is contained in BCC4 and MORN1 that form a 'rubber band' to sequester the basal end of the assembling daughter cytoskeleton. Finally, we assign BCC1 to the non-essential, final BC constriction step.

The Apicomplexa are obligate intracellular parasites that can infect a wide range of hosts. *Toxoplasma gondii* is widespread and has infected one-third of the global human population. Although most infections are chronically dormant and symptomless, opportunistic infections lead to a spectrum of clinical manifestations in the immunocompromised, immunosuppressed, or individuals with an immature immune system[1]. All pathology originates from tissue destructions caused by fast rounds of lytic, intracellular cell divisions of the acute, tachyzoite life stage.

*T. gondii* cell division differs in many respects from the mammalian cell division conventions. Tachyzoites divide by budding two daughter cells inside the mother cell (i.e., endodyogeny = internal budding). In this process, daughter cytoskeleton scaffolds nucleate on the duplicated centrosomes and grow in an apical-to-basal direction[2–5]. Their cortical membrane skeleton is principally different from the actin-spectrin cytoskeleton in mammals and imposes distinct needs on

the cell division machinery[6]. Many of the cytoskeleton components are unique to the parasite and absent from the mammalian host, such as a family of intermediate filament-like proteins associated with alveolar vesicles making up the inner membrane complex (IMC)[5,7]. The membrane skeleton is buttressed by 22 sub-pellicular microtubules emanating from the apical end, which itself is capped by a unique microtubular basket known as the conoid.

A ring structure situated on the very basal, posterior edge of the daughter scaffolds is known as the basal complex (BC) and is essential to complete cell division[8–11]. Surprisingly, although the BC functions as the contractile ring during cell division[8,11], *T. gondii* can complete cell division in absence of actin[12,13], unlike the archetype actin-dependent contractile ring. Consistent with this, although Myosin J (MyoJ) is responsible for the last step of BC constriction, parasites lacking MyoJ only display mild fitness loss[14]. However, interfering with BC assembly at earlier steps induces lethal phenotypes, either by overexpressing

[1]Department of Biology, Boston College, Chestnut Hill, MA, USA. [2]Department of Chemistry, Boston College, Chestnut Hill, MA, USA.
✉e-mail: engelbek@bc.edu; gubbelsj@bc.edu

the BC scaffolding protein MORN1[8,15], or through MORN1 depletion, results in daughter scaffolds that are much wider open and leads to conjoined, double- or multi-headed parasites[11]. Thus, the early assembly of the BC is essential to complete cell division, which implies there is an essential, actin-myosin-independent process acting early in cell division.

The inventory of proteins thus far mapped to the BC does not immediately highlight the unconventional mechanism of this essential function during cell division. The parts known so far have provided insights into the architecture and spatiotemporal dynamics of the BC. Three to four groups of spatiotemporally defined IMC proteins are sequentially recruited and serve as a set of highly-resolved daughter development markers[5,7]. Scaffolding protein MORN1 is deposited already at formation of the daughter bud[8]. Halfway through daughter development, additional proteins are recruited to the BC such as the intermediate filament-like IMC5, IMC8, IMC9, and IMC13[7] as well as MyoJ[14] and Centrin2 (Cen2)[9,16]. This recruitment coincides with the onset of tapering the cytoskeleton scaffolds to the posterior end[7,16]. A last transition in BC composition occurs upon daughter parasite emergence when proteins such as FIKK[17] and MSC1a[15] are recruited. The role of these proteins, which are only found in the BC of the mother cell, is currently not well understood[18]. The set of known BC markers in

combination with electron microscopy resolves into three sub-structures within the BC[7,16,19] (Fig. 1a). It is of note that both MORN1 and Cen2 have additional localizations in the cell: MORN1 in the spindle pole (centrocone) and apical end of the cytoskeleton[8] and Cen2 in the centrosome, apical annuli and apical polar ring[9].

Here, we analyze how the BC is essential for cell division by application of reciprocal BioID using six different baits. Statistical analysis and experimental validation identifies 11 novel BC components (BCC1-11) resolving across four BC sub-clusters (BCSC1-4) correlating with the known architecture[7]. In addition, we identify a protein preceding BC formation (BCC0) that lays out a 5-fold symmetry conserved throughout the cytoskeleton in the apical annuli and alveolar vesicle architecture. Our data show that the daughters bud bi-directionally from this structure. Furthermore, we discover that BCC4 depletion phenocopies MORN1 depletion, and while not essential for BC formation, it is required to maintain the structural integrity of nascent daughter buds. This leads us to propose a 'rubber band' model that mechanistically explains the essential role of the BC in early stages of cell division. Overall, our work uncovers the early basis of the 5-fold symmetry in cytoskeleton architecture, adds another dimension to the hierarchy and structure of daughter budding, and answered the long-standing question of the essential function of the BC during cell division.

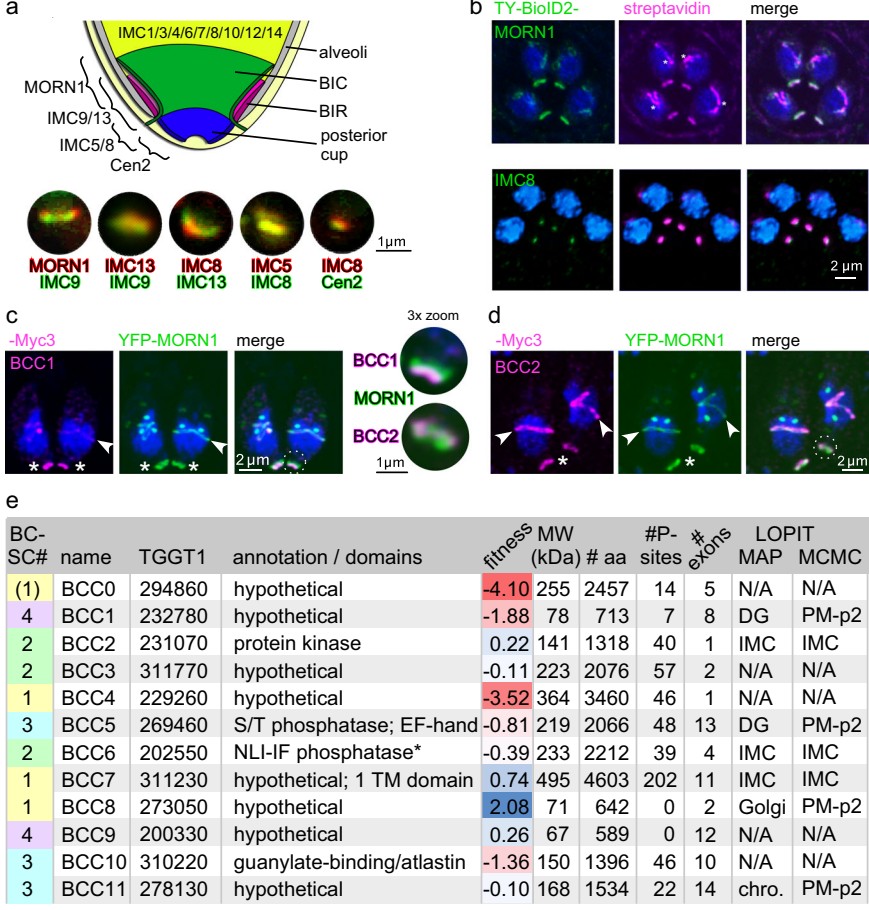

**Fig. 1 | Proximity biotinylation reveals additional BC components. a** Schematic of the *T. gondii* BC highlighting distinct localizations of the utilized BioID-baits. Drawing and double-stained panels adapted from ref. 7. **b** Endogenously tagged BioID2 parasite lines in the MORN1 or IMC8 locus, show correct localization and biotinylation capability in presence of 150 µM biotin for 16 hrs. **c, d** Examples of subcellular localization analysis of endogenously C-terminal 3xMyc-tagged BCCs co-stained with YFP-MORN1 to confirm BC localization. Distinct temporal localization kinetics to the mother (asterisks) and daughter BC (arrowheads) are

revealed, as well as distinct spatial patterns relative to MORN1 in the mature BC (3x zoom panels of the regions marked with the dotted lines). **e** All proteins mapped to the BC in this study. BCSC assignments based on averaged statistical analysis combined with experimental validation. Data derived from ToxoDB[27], which harbors the primary data reported as follows: phosphoproteome[29], lytic cycle fitness score[28], hyperLOPIT subcellular localization assignments[30]. * Nuclear LIM Interacting factor family phosphatase.

## Results

### Mapping the basal complex by reciprocal proximity biotinylation

We previously defined distinct compartments in the BC[7] and started with four baits representing these compartments (MORN1 representing the widest upper part, IMC8 the middle section, and Cen2 and MyoJ both representing the lower section of the BC) for mapping the BC proteome by proximity biotinylation (Fig. 1a). We fused the endogenous ORFs at their 5′-end with Ty-tagged BioID2 and confirmed the correct localization and biotinylation capacity by fluorescence microscopy (examples for MORN1 and IMC8 shown in Fig. 1b; Ty-BioID2-Cen2 was reported before[20]). Following mass spectrometry, we first assured that the biological replicates were of high reproducibility (Supplementary Fig. 1) and subsequently mined the data for novel BC components (BCCs). As described in detail below, we analyzed the mass spectrometry data using the Significance Analysis of INTeractome (SAINTexpress) algorithm[21,22], designed to identify interaction partners in AP/MS or proximity-biotinylation applications. We experimentally validated BCC components out of these initial experiments (Fig. 1c, d, Supplementary Fig. 2; BCCs were numbered in the order in which we validated them), and selected BCC1 and BCC2 for a reciprocal round of BioID2 using the same strategy as above. SAINTexpress analysis of all data resulted in successful recovery of many known BC proteins, whereas upon further experimental validations we identified an additional set of 12 novel BC-associated proteins (summarized in Fig. 1e) as described below in detail.

### Assembly of the BC map

To extract architectural insights from our proximity labeling, significantly enriched preys from all six baits were used to generate prey-prey maps by correlating the abundance of individual preys over all biotinylation experiments[23,24]. This permitted mapping of potential colocalization and visualized subcellular complexes (Fig. 2, Supplementary Fig. 3, Supplementary Data 1). We ran our analyses with or without the inclusion of a cytoplasmic BioID2-YFP fusion control[20] (Supplementary Fig. 3a vs Fig. 2a), and with various FDR and abundance cut-off settings to reduce the false-positive detection of spurious cytoplasmic proteins, while keeping all known BC components in the dataset. Throughout this multi-round, iterative process we tagged putative BCC proteins, which led to the serendipitous mapping of the apical annuli[20], as well as identified several new proteins localizing to the very apical end of the parasites, as well as the spindle's centrocone (Supplementary Fig. 4, Supplementary Data 1). These known and unknown non-BC hits stemmed predominantly from the non-exclusive BC baits MORN1 and Cen2 as well as from IMC8, which initially is deposited on the whole daughter scaffold before transitioning to the BC, halfway through cell division. Determining the BC proteome was further complicated by the differential temporal association of different BC proteins with the BC as well as proteins that are initially recruited to the BC during division but upon maturation end up in other structures like the apical annuli and the alveolar sutures (proximity biotinylation occurs at all times and structures). The prey-prey map comprising all newly mapped BCCs is provided in Fig. 2a with select details of the statistical analysis resolved per bait provided in Fig. 2b, whereas more extensive analyses are included in Supplementary Fig. 3 and Supplementary Data 1. Note that all these analyses include parts of the additional MORN1/Cen2/IMC8 proteomes, which could only be untangled by extensive experimental validation.

The prey-prey distance heat maps consistently revealed four major clusters. These clusters, which we named BC Sub-Clusters 1–4 (BCSC1-4; Fig. 2a, Supplementary Fig. 3a), confirmed our previous observations of the BC architecture that were anchored on MORN1 (BCSC-1), IMC9/13 (BCSC-2), IMC5/8 (BCSC-3) and Cen2 (BCSC-4)[7]. As eluded to above, several additional subcellular localizations besides the BC were interspersed in these clusters, including the set of apical

annuli proteins (AAPs) for Cen2[20] and centrocone proteins (e.g., TgCENP-C (ToxoDB user comment), TGGT1_265770, TGGT1_258450, and TGGT1_270810 (Supplementary Fig. 4)) for MORN1 (Supplementary Data 1). Therefore, assigning uncharacterized components in these clusters to the BC substructures through 'guilty-by-association' strategy in the prey-prey maps was not reliable. Further comparison of prey abundance between bait samples, however, added higher resolution as represented in the dot-plots (Fig. 2b, Supplementary Fig. 3b), although this analysis is slightly underpowered by using only two biological replicates (each with two technical MS replicates). Especially candidates exclusively detected with Cen2 or MORN1 could in general be triaged for presence in the BC (Supplementary Fig. 3b). With this in mind, we can with reasonable confidence assign hypothetical protein TGGT1_225270 to BCSC-1, and CDPK6 to BCSC-3.

In addition, proteins not present in the BC do provide insights in how the BC might be interfacing with the IMC, e.g., IMC29 and suture proteins TSC3 and 4[25] suggest the BC as site of daughter bud growth (e.g., palmitoyl transferase DHHC14, which is critical for IMC assembly[26]) (Fig. 2b, Supplementary Data 1). The detection of this set of proteins across 2–4 BC baits, notably IMC8, supports that these interactions are not spurious (Fig. 2b, Supplementary Fig. 3b). Overall, reciprocal BioID of the BC aligned with the physical BC substructures previously detected, identified associations between the BC and IMC structures, and assigned several previously uncharacterized hypothetical proteins to different BC substructures.

### Identified BCCs comprise a diverse set of proteins

We characterized the new BCCs for putative function by mining ToxoDB[27] for the following data: lytic cycle fitness score[28], phosphoproteome[29], hyperLOPIT subcellular localization assignments[30], and functional (domain) annotation (Fig. 1e). This revealed 8 hypothetical protein BCCs, lacking known domains or function. BCC2 contains a kinase domain, whereas BCC5 and BCC6 are phosphatases. Phosphatase BCC5 harbors an EF-hand, indicating this protein likely binds calcium, whereas BCC6 is annotated as a nuclear LIM interacting factor (NLI-IF) family phosphatase. NLI phosphates typically dephosphorylate the RNA Polymerase II CTD, however, much more divergent functions are found in many *Arabidopsis thaliana* NLI phosphatases[31]. BCC10 harbors a guanylate-binding domain and a weak atlastin homology, which in vertebrates is a dynamin-like GTPase required for fusion of endoplasmic membrane tubules[32]. Finally, BCC7 harbors a TM domain, whereas BCC0 contains predicted myristoylation and palmitoylation sites, which makes these the only two putatively membrane-anchored BCCs. Of all novel BCCs, only BCC0 and BCC4 had fitness scores suggesting a potential essential role (score <−2)[28].

Next, we performed a global analysis of distinct BC spatiotemporal localization kinetics for the various BCCs (Supplementary Fig. 2). This revealed that BCC6 and 7 exclusively localize to the BC of mature parasites, following completion of cell division (like FIKK[17] and MSC1a[15]), whereas BCC3 and 4 exclusively localize to the budding daughter BC, however, BCC3 presents a special case as it localizes to the whole daughter bud, which prioritized it for follow up. All other BCCs localize to the BC of both the budding daughters and mature mother. Combining all these insights for their unique features, we selected BCC0, 1, 3, and 4 for more detailed experimental exploration, presented below in order of appearance at the BC.

### BCC0 provides a foundation for the BC and other cytoskeletal elements

The severe −4.10 fitness score suggests BCC0 is essential, but besides predicted myristylation and palmitoylation sites, the primary structure does not provide clues toward its function (Fig. 3a). C-terminal tagging of BCC0 with a spaghetti monster (sm)-Myc-tag revealed its association with the centrosomes immediately after their duplication. As the

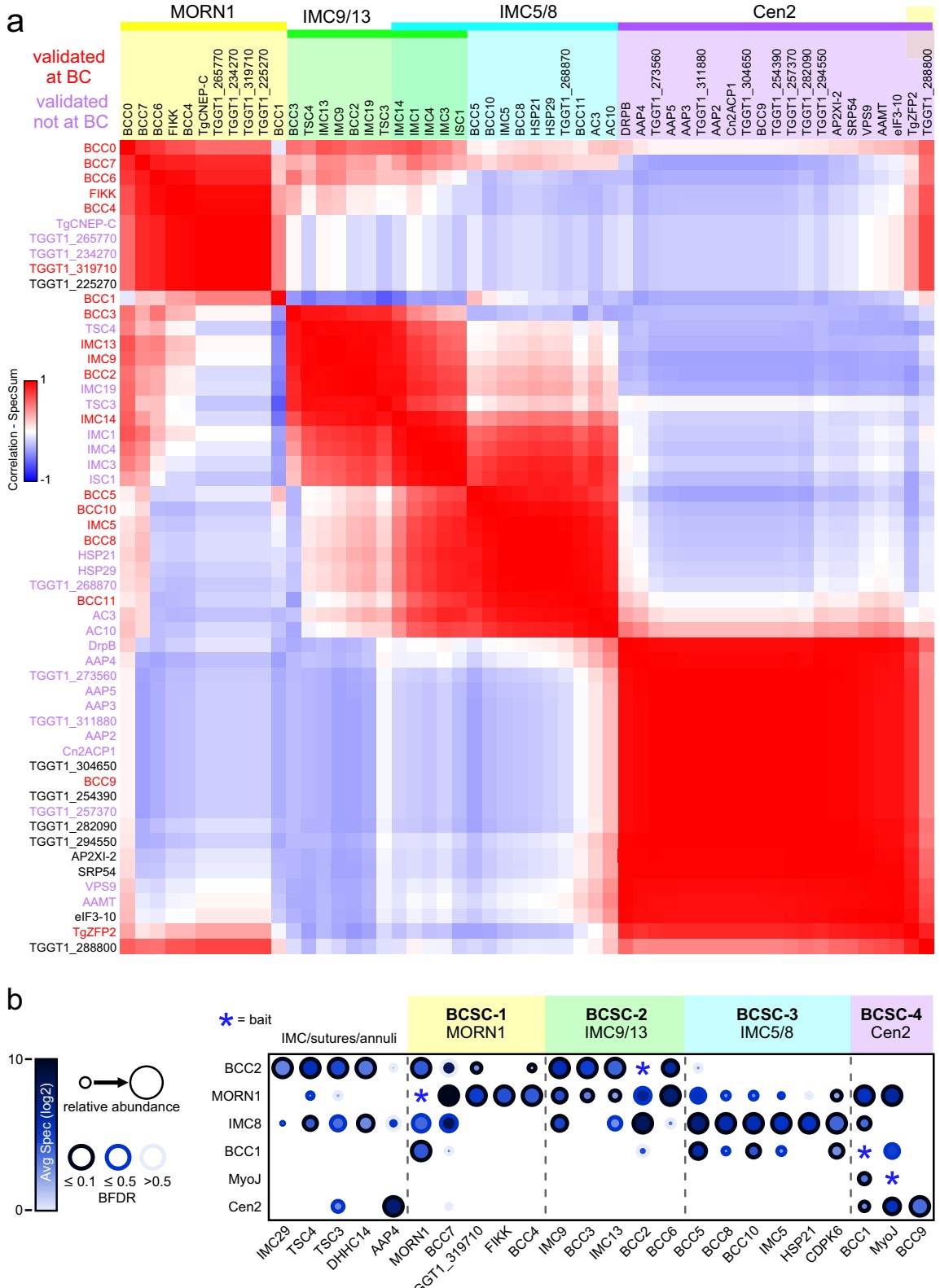

centrosomes separate, the signal condenses into several foci surrounding the centrosomes (Fig. 3b; additional cell division cartoons present below and in Supplementary Fig. 6). Next, we determined that BCC0 is recruited to the daughter scaffold before ISP1, a palmitoylated IMC component localizing to the apical cap alveolus that is one of the earliest bud markers[33] (Fig. 3c). As expected, ISP1 first shows up with moderate intensity before the signal intensifies and extends across the

apical cap. Notably, BCC0 is already visible as two clusters of foci marking early daughter scaffold assembly before ISP1 is detected (Fig. 3c, left panels). Therefore, BCC0 is an earlier marker of daughter budding than ISP1. An additional observation is that at the time of apical cap formation as marked by ISP1, BCC0 has already started to extend in the basal direction away from ISP1 (Fig. 3c, middle panels). This strongly suggests that the apical cap is assembled in an apical

**Fig. 2 | Reconstruction of the BC architecture via spatial relation analysis.**
**a** Distance heatmap visualizing the BC proximity landscape, partitioned into four major BC subcomplexes (BCSCs) marked in yellow (BCSC-1), green (BCSC-1) (BCSC-1), blue (BCSC-1), and purple (BCSC-1). Proteins with overlapping partitioning to two different BCSCs are dual-colored accordingly. The defining and localization validated components are named at the top of each BCSC colored group. The distance heatmap was visualized with the following settings: abundance column set to "average spectral counts", score column set to "false discovery rate (FDR)", score filter = 0.1, secondary score filter = 0.25, lower abundance cutoff for prey correlation = "10 spectral counts". Prey order is the same for the x- and the y-axis. See Supplementary Fig. 3 for inclusion of a different control resulting in looser cluster definition. **b** Dot plot summarizing known and newly identified BCCs over the executed experiments, including the BC proteins used as BioID2 fusion baits. Preys that pass an initial filter of FDR ≤ 0.1 for at least one bait, were visualized and manually sorted in the four BCSCs as defined in panel A (BCSC-1 assignments were based on panel 'a' as well as additional analyses using different settings and controls; see Supplementary Fig. 3). Further criteria were: abundance column set to "average Spectral Counts", secondary score filter set to "false discovery rate (FDR) ≤ "0.5" and the log transformation = "log2". The IMC/sutures/annuli group defines a separate group in this analysis, which shares overlap connections mostly with BCSC-2 and to a lesser extend with BCSC-3. The entire dot plot is shown in Supplementary Fig. 3b. Source data are provided as a Source data file.

direction. Moreover, the basally migrating BCC0 has a 'beads on a string' appearance (Fig. 3c, right panels), a pattern reminiscent of the longitudinal sutures between the alveolar plates[25,34,35].

None of the BCC0 patterns are consistent with a BC localization. To investigate the connection of BCC0 with the BC we visualized MORN1 by an endogenous YFP fusion and performed SR-SIM microscopy. This revealed a sharply defined pattern of in general five BCC0 foci laying right on top of the early BC (Fig. 3d, Supplementary Movie 1). Early BCC0 appearance in daughter development is communicated by the barely separated MORN1 signals in the centrocones, demonstrating that the spindle poles just separated (Fig. 3d cartoon). Next, we asked whether these five foci could represent (the basis) of the apical annuli, which display a similar five-fold symmetry at the suture separating the apical cap from the rest of the IMC[20]. Neither during initiation nor at a later time point of development could we appreciate significant co-localization of apical annuli marker AAP4 and BCC0 (Fig. 3e; note aspecific AAP4 antiserum cross-reaction with the centrosome[20]). This suggests they might not be structurally related, although we cannot exclude the possibility that BCC0 provides the foundation for the annuli, which do not appear till later in cell division[20] at which point BCC0 appears to already have extended along the longitudinal sutures. Overall, BCC0 seems to lay down an early 5-fold symmetry transitioning into the sutures, and is potentially set to form the spatial que for BC formation. Its appearance preceding BC formation inspired the name BCC0 for this protein.

To dissect BCC0 function we replaced its promoter with a tetracycline-regulatable promoter[36] (Supplementary Fig. 5a). BCC0 is critical for completing the lytic cycle (Fig. 3f) and is depleted upon ATc addition (Fig. 3g). Although the daughter IMC scaffold initially looks fairly normal, the deposition of both apical annuli and MORN1 in the forming daughter buds is compromised (Fig. 3g–i). Prolonged BCC0 depletion results in large cytoplasmic masses (plasma membrane marker SAG1), loss of IMC shape, and the appearance of nuclei not encased by IMC (Fig. 3j). Overall, the data suggest that BCC0 is important for depositing MORN1 in the BC as well as positioning the annuli; although IMC morphology is initially normal, lacking these key structures leads to an unstable, aberrant IMC.

### BCC3 dynamically localizes to the bud initiation complex, the sutures, and the BC

BCC3 was selected for its exclusive association with the daughter cytoskeleton. Besides a coiled-coil domain BCC3 displays no functional features (Fig. 4a). We tagged BCC3 at the C-terminus with a triple Myc-tag and imaged it throughout parasite division using IMC3 as bud marker. BCC3 also appears as five foci early in cell division just before IMC3 shows up at the daughters. Sequentially throughout daughter budding, BCC3 is initially seen in a pattern reminiscent of the longitudinal sutures, whereas mid-budding an additional signal forms at the suture below the apical cap, while toward the end of budding the signal exclusively localizes to the daughter BCs before completely disappearing upon conclusion of cell division (Fig. 4b). Co-staining

with YFP-tagged MORN1 showed that besides co-localizing in the BC, BCC3 sits right below MORN1 at the apical end of the IMC in the forming buds (Fig. 4c, blue arrowheads). Co-staining with AAP4-showed that BCC3 localization was bordered by the apical annuli and did not extend into the apical cap. Since BCC3 has a modest fitness score we did not attempt a knock-out. In conclusion, BCC3 is a very dynamic daughter bud marker present at its foundation as five foci and then transitions onto the daughter scaffold sutures before accumulating on the BC, from where it is released upon completion of cell division.

### BCC4 exclusively associates with the budding daughter BC

Standing out for its severe fitness score, BCC4 is a hypothetical protein harboring a single coiled-coil domain (Fig. 5a). We tracked BCC4 (endogenously tagged with 3xMyc) throughout cell division using β-tubulin as co-stained guide. BCC4 appears early in the division cycle in close proximity to the spindle microtubules and is associated with early daughter bud formation (Fig. 5b). For the remainder of division BCC4 is present at the BC of nascent buds, but disappears upon emergence of daughter cells. To further pinpoint the early temporal events, we colocalized BCC4 with the centrosome. BCC4 accumulated on top of the just duplicated centrosomes (Fig. 5c, 1) and progressively condenses into rings around the centrosomes during S/M-phase (Fig. 5c, 2–3). Since the early BC formation is mediated by MORN1, which exhibits a similar dynamic and is first visible as clouds around the divided centrosomes that subsequently form rings[3,8,9,16], we co-colocalized all three together. This demonstrated that BCC4 and MORN1 simultaneously accumulate distal to the centrosome (Fig. 5d) and subsequently transform into the ring-shaped BC (Fig. 5e). However, in contrast to MORN1, BCC4 exclusively localizes to the BC and is released from the BC when cell division completes (Fig. 5e, arrowheads), thereby sharply focusing its role in cell division.

### BCC4 is an essential BC component

To test the function of BCC4 we replaced its promoter with a tetracycline-regulatable promoter. Plaque assays demonstrate that BCC4 is essential for in vitro proliferation (Fig. 6a). Phenotype analysis using AAP4 and IMC3 markers showed that BCC4 depletion results in the formation of double-headed parasites conjoined at their basal end (Fig. 6b). The same phenotype was also seen upon depletion of MORN1, which was the result of a defect in BC assembly[11], as well as upon depletion of phosphatase HAD2a[37].

Since promoter-replacement acts on gene transcription and relatively slowly permeates into protein kinetics, we also fused BCC4 to the auxin-inducible degron (AID) system to deplete BCC4 at a much faster rate[38] (Supplementary Fig. 5b). To enable direct comparisons, we generated a Ty-mAID-MORN1 line as well (Supplementary Fig. 5c). BCC4-mAID-3xMyc or Ty-mAID-MORN1 parasites exhibited small ~7 times smaller plaques after 7 days of IAA treatment compared to the parental or non-induced populations (Fig. 6c). This indicates that in contrast to the ATc regulated BCC4, parasite proliferation is severely

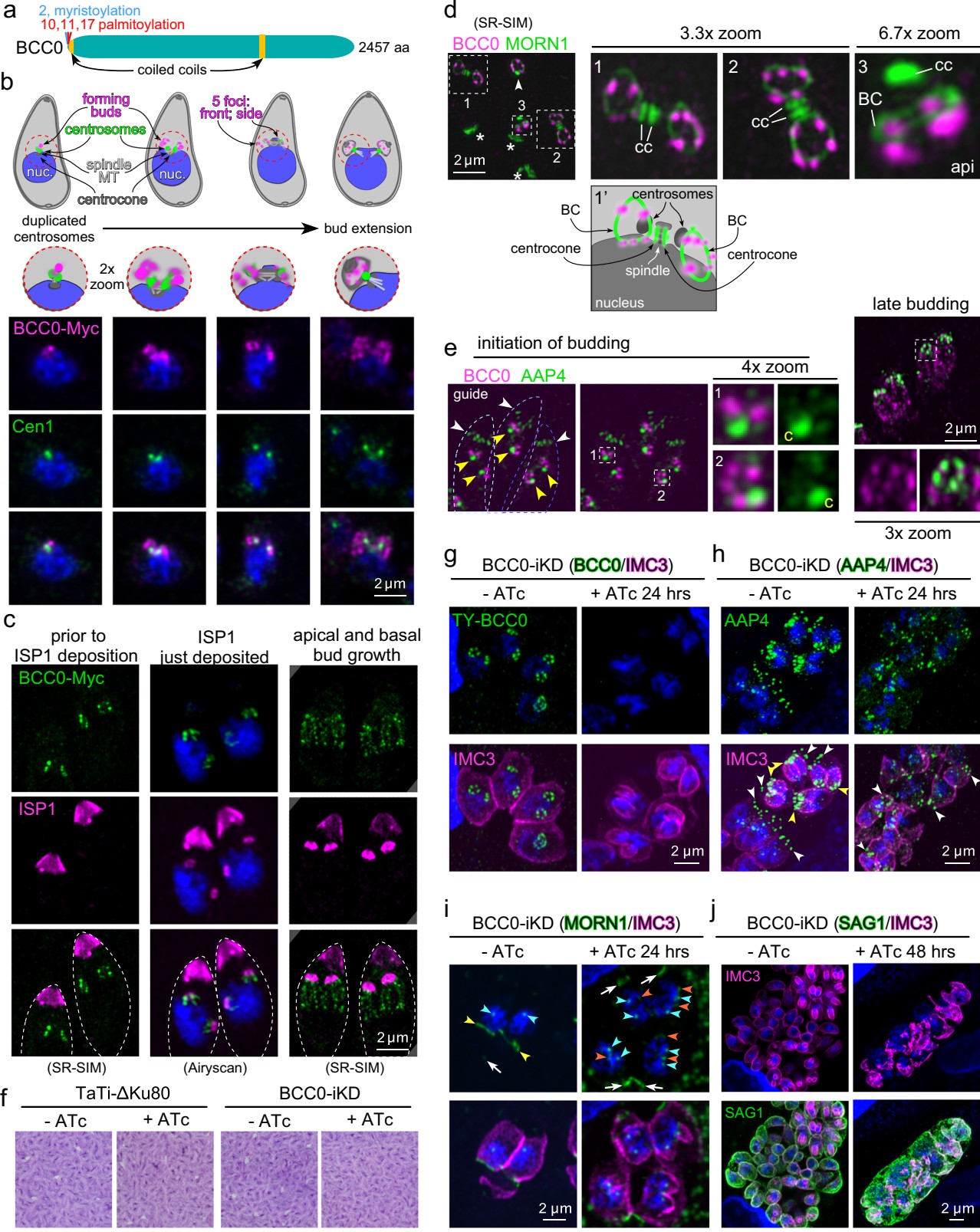

inhibited but not completely disrupted by the AID system (Fig. 6d). Interestingly, after 14 days of growth the size of auxin treated Ty-mAID-MORN1 plaques significantly increases, which is not seen for the BCC4-mAID parasites (Fig. 6d). For both the MORN1 and BCC4 mAID lines, the proteins are largely depleted after 2 hrs of IAA treatment, which already resulted in wider basal ends of the parasite buds (Fig. 6e). To follow BC fate after BCC4 or MORN1 degradation, we reciprocally tagged BCC4 and MORN1 in the mAID parasite lines. Although, 2 hrs of MORN1 or BCC4 depletion did not abrogate recruitment of its counterpart (Fig. 6f), we observed wide daughter buds in both scenarios. Measurements of the daughter basal end diameters showed the widths were similar in the BCC4 and MORN1 depleted parasites, which were significantly wider than detected in controls (Fig. 6g). In summary, BCC4 phenocopies MORN1.

**Fig. 3 | BCC0 is a component of the nascent daughter bud. a** Schematic representation of BCC0 protein features. **b** Endogenously BCC0 tagged with spaghetti monster Myc tag (smMyc) co-stained for the centrosome (Centrin antiserum). DNA is stained with DAPI (blue) and applies to all panels with blue stain. The cartoons provide annotation and whole parasite perspective. **c** BCC0-smMyc colocalization with ISP1 antiserum. BCC0 at the daughter bud precedes ISP1. When ISP1 is deposited, BCC0 starts to elongate basally. Dotted lines outline parasites. **d** SR-SIM of BCC0-smMyc co-expressing endogenously tagged YFP-MORN1. Upon MORN1 appearance, BCC0 is present as 5–6 dots. Boxed regions in left panel correspond with the zoom panels. Asterisks mark mother BC; arrowhead marks the other centrosome and daughter bud in the middle parasite corresponding with box #3; 'cc' marks the centrocones (spindle poles); 'api' marks the apical end of this bud. Cartoon 1' provides annotation of panel 1. Supplementary Movie 1 provides a 3D rendered rotation. **e** Airyscan images of BCC0-smMyc parasites co-stained with AAP4 antiserum (apical annuli). The 'guide' panel outlines 3 parasites in dotted lines

(shades of blue); white arrowheads mark mother's apical annuli; yellow arrowheads or 'c' mark centrosomes. **f** Plaque assays of inducible knock-down of BCC0 (BCC0-iKD) demonstrate that BCC0 is essential. The endogenous promoter of BCC0 was replaced with a TetO7sag4 anhydrous tetracycline (ATc) regulatable promoter fused to a Ty tag. TaTi-ΔKu80 is the parent line. Representative of $n = 3$ biological replicates. **g** BCC0-iKD parasites co-stained with Ty (BCC0) and IMC3 (IMC scaffolds) antisera. BCC0 depletion results in disorganized daughter IMC. **h** BCC0-iKD parasites co-stained with AAP4 and IMC3 antisera. BCC0 depletion misorganizes daughter annuli. White and yellow arrowheads mark maternal and daughter apical annuli, respectively. **i** BCC0-iKD parasites co-stained with endogenously expressed YFP-MORN1. White arrows and yellow arrowheads mark maternal and daughter BCs, respectively; orange arrowheads mark partial/broken daughter BC. Teal arrowheads mark MORN1 in the centrocone. **j** BCC0-iKD parasites co-stained with SAG1 (plasma membrane) and IMC3 antisera. Parasites continue expansion, but lose their shape. Some parasite nuclei are not encased by IMC.

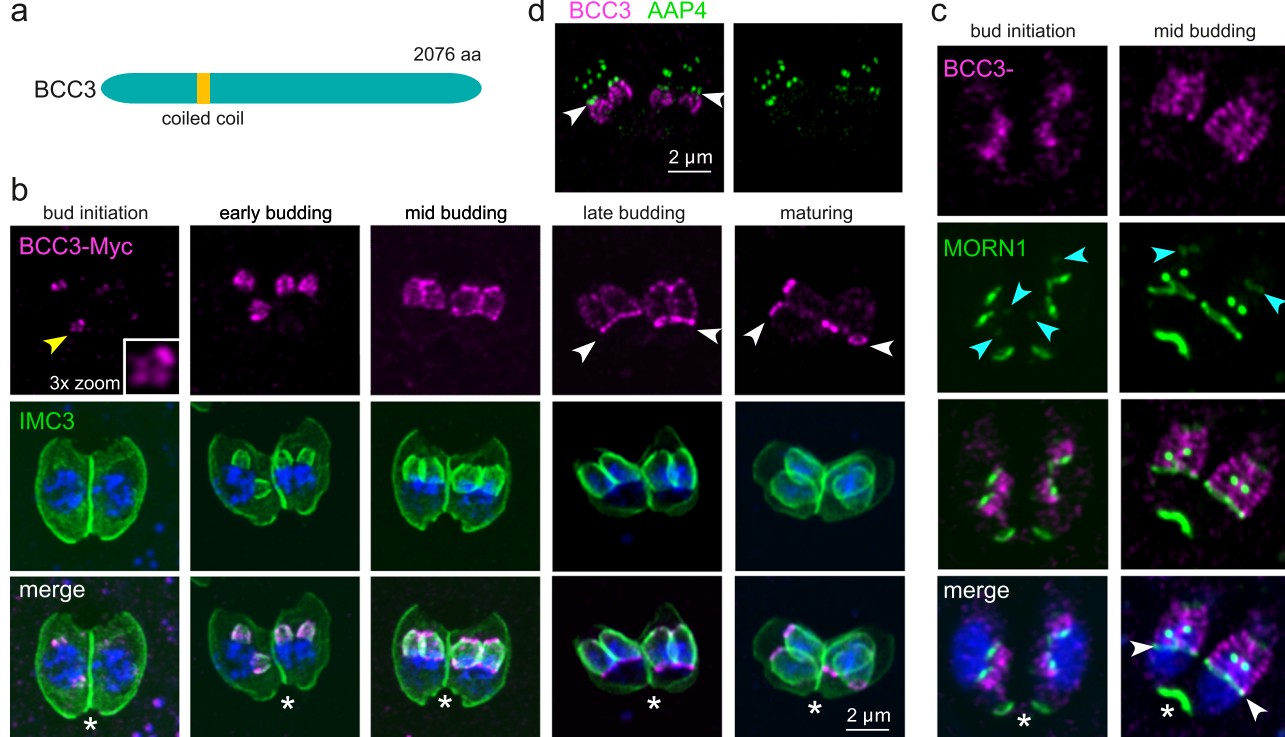

**Fig. 4 | BCC3 dynamically localizes on the IMC scaffolds and BC during daughter budding. a** Schematic representation of BCC3, which only harbors a single coiled-coil region as recognizable domain. **b** Parasites expressing endogenously C-terminally Myc3-tagged BCC3 were co-stained with IMC3 antiserum, which shows that BCC3 localizes to the 5-foci early in cell division (yellow arrowhead, enlarged in box) and then extends basally along the developing cytoskeleton in a speckled pattern through the first half of budding. Upon maturation of the daughters, BCC3 transitions completely to the daughter BC (arrowheads) and eventually releases from the cytoskeleton in mature parasites (asterisks). DNA is stained with DAPI (blue) and applies to all panels with blue stain. **c** Co-stain of Myc3-tagged BCC3 with endogenously tagged YFP-MORN1. Blue arrowheads mark

MORN1 at the apical end of the IMC, and the BCC3 signal extends almost to this signal in the early stages, but at mid-budding BCC3 ends more basally in a thick line, likely at the suture below the apical cap (see the same stage in panel b for a comparable signal). Note the localization of BCC3 to the daughter bud IMC sutures during the mid-steps of division and the prominent co-localization with MORN1 in the daughter BC later in division (white arrowheads) but BCC3 is absent from the BC of the mother cell (asterisks). Scale bars are the same as shown in panel b. **d** Co-stain of 3xMyc-tagged BCC3 with AAP4 antiserum to mark the apical annuli in an early-mid stage budding parasite demonstrates that BCC3 is not present in the apical cap and suggests the apical cap grows apically at the same time as the more basal alveoli grow in the basal direction.

## BCC4 is required for maintaining BC integrity throughout cell division

We next analyzed the detailed kinetics of the double-headed phenotype (Fig. 7a vs Fig. 6b). We first ascertained the double-headed parasites in the BCC4-mAID-3xMyc and Ty-mAID-MORN1 lines indeed had two apical ends by AAP4 staining (Fig. 7b). Double-headed parasites enumeration revealed that BCC4 depleted parasites show a nearly 2-fold higher incidence of more than two double-headed daughter parasites per vacuole compared to MORN1 depleted parasites (Fig. 7c).

This more penetrant BCC4 phenotype is consistent with the plaque assay results (Fig. 6c).

To dynamically track the BC upon BCC4 depletion, we performed time-lapse microscopy of BCC4-mAID parasites co-expressing endogenously YFP-tagged MORN1. We started imaging vacuoles containing two or four parasites in division when nascent MORN1 rings were visibly emerging around centrocone-localized MORN1. In untreated parasites the MORN1 signal at the apical end of nascent daughter cells was detected at $t = 24$–$30$ min and displayed the basal to apical

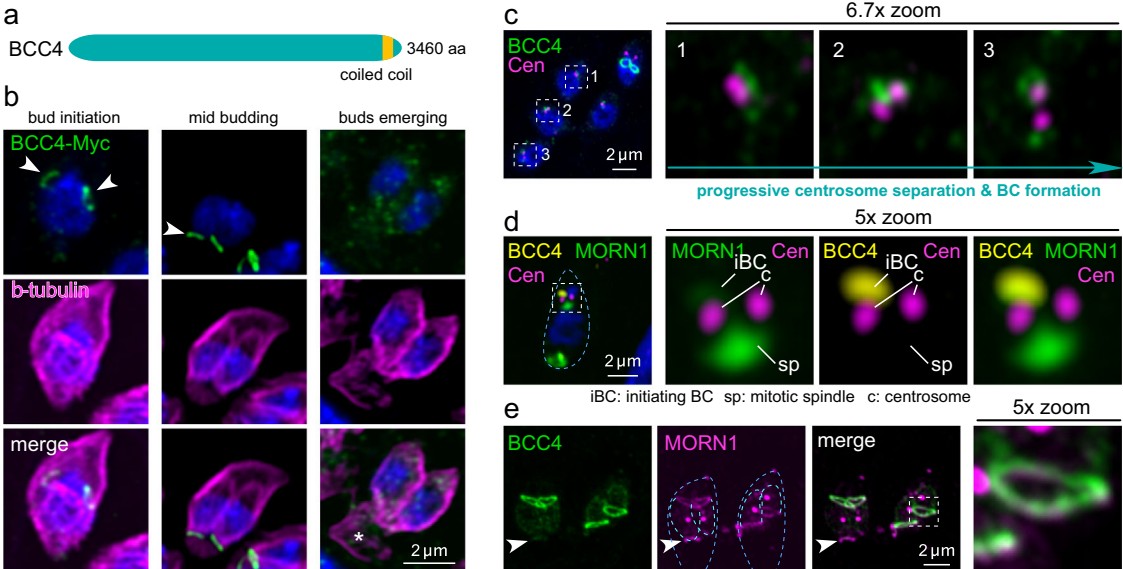

**Fig. 5 | BCC4 only localizes to the BC during division. a** Schematic representation of BCC4, which only harbors a single coiled-coil region as recognizable domains. **b** Parasites expressing endogenously C-terminally Myc3-tagged BCC4 were co-stained with β-tubulin antiserum, which shows that BCC4 localizes to the BC right at its formation but is absent from the mature BC. Arrowheads mark the early- and mid-development daughter BCs; asterisk marks the retracting and disassembling mother's cytoskeleton. DNA is stained with DAPI (blue) and applies to all panels with blue stain. **c** Co-staining of BCC4-Myc parasites and the *T. gondii* centrosome (*Hs*Centrin2 antiserum) shows that BCC4 assembles as a ring-like structure at the point of centrosome duplication. The boxed and numbered panels are magnified as

indicated and show BCC4 transitioning from an undefined bleb around the centrosome in panels 1 and 2 into a ring as visible in panel 3. **d** Four color imaging displays the interplay of YFP-MORN1 and BCC4-Myc3 and the centrosomes in early BC formation. At this very early stage of daughter development before completion of spindle pole separation, BCC4 and MORN1 co-localize in an amorphous mass in close apposition to the outer centrosome. Blue dotted line outlines the parasite. **e** YFP-BCC4 co-localizes with 5xV5-MORN1 in the BC around the midpoint of daughter cell budding, but in contrast to MORN1, BCC4 is absent from the mature BC in the mother parasite (arrowhead). Dark blue dotted lines outline the mothers; light blue dotted lines outline the budding daughters.

expansion of daughters (Fig. 7d open arrowheads; Supplementary Movie 2). From here, the BC initially expanded until approximately minute 54 and then moves with the basal end of forming daughters, surpassing the centrocone between 60 and 66 min (Fig. 7d closed arrowheads). Contrary to these dynamics, parasites to which IAA was added 2 hrs before *t* = 0 failed to produce visible daughter bud apical ends by minute 45. The daughter BCs formed close to the centrocones as seen in controls and showed an initial expansion (Fig. 7d blue arrowheads; Supplementary Movie 3). However, at 54 min two of the four BCs started to fragment, and by 75 min all are broken up (red arrowheads). Thus, in absence of BCC4, MORN1 rings initially form, but midway through daughter assembly the BC falls apart. The critical loss of MORN1 at this point prevents tapering of the daughters during the progression of division[10,11].

To gain higher resolution information of the BC fragmentation and investigate how the BC interfaces with the microtubular cytoskeleton we applied ultrastructural expansion microscopy (U-ExM)[39]. BCC4-mAID-3xMyc parasites co-expressing YFP-MORN1 were co-stained with β-tubulin. Upon 24 hrs IAA induction the sub-pellicular microtubules appeared frayed at the daughter buds' basal ends (Fig. 7e vs. Fig. 6e, f). Although MORN1 still localizes to the basal end of the sub-pellicular microtubules, it lost its smooth and continuous ring-like appearance as seen in untreated parasites, suggesting that loss of BC integrity abolished the bundling of the (+)-ends of the sub-pellicular microtubules (Fig. 7e).

The continued association of MORN1 with the daughter microtubule (+)-ends also hints at a mechanism for positioning the BC at the daughter buds. To test this, we overexpressed MORN1, which induces formation of MORN1 rings and disrupts IMC formation[8]. However, sub-pellicular microtubule cytoskeleton formation remains unaffected, though the microtubule ends are no longer bundled and frayed. When we transiently overexpressed mCherryRFP-MORN1 in BCC4-3xMyc parasites, MORN1 accumulated in three distinct spots corresponding

with mother and daughter BCs (Fig. 7f). Under these conditions, BCC4 did not co-localize with MORN1 and failed to assemble in rings. In contrast, BCC4 was dispersed over the entire parasite and foci were present along the sub-pellicular microtubules, but not restricted to the BC (Fig. 7f).

In conclusion, our data support complementary roles for BCC4 and MORN1 in maintaining BC integrity beyond the budding midpoint when MyoJ and Cen2 are recruited. However, in contrast to MORN1, BCC4 is only temporally needed till division is completed. An essential role of the BC is therefore the bundling of subpellicular microtubule ends. These data highlight a critical new protein and event in the BC.

### BCC1 is essential for final tapering and recruits MyoJ and Cen2 to the BC

BCC1 is present in BCSC-4 together with MyoJ and Cen2 (Fig. 2b) suggesting a potential role in the final BC constriction. Coiled-coils are the only identifiable feature, which do not permit a prediction of BCC1 function (Fig. 8a). Indeed, like MyoJ and Cen2, tagged BCC1 is recruited to the BC at the budding midpoint (Fig. 8b), and co-localizes robustly with them but not with MORN1 in the posterior cup in non-dividing parasites (Fig. 8c).

BCC1 has a moderate fitness score of −1.88[28], which we tested by placing it under the tetracycline-regulatable promoter and simultaneous insertion of a Ty-tag (Supplementary Fig. 5d). No difference in plaque size was observed, indicating that BCC1 is not essential (Fig. 8d), which mirrors MyoJ[14]. BCC1 depletion prevented association of Cen2 and MyoJ with the BC, but did not affect other subcellular localizations of Cen2 (Fig. 8e). Similar to depletion of MyoJ, Cen2, or actin, which all cause defects in final BC constriction and likely the formation of the posterior cup[10,13,14], BCC1-depleted parasites appeared 'stumpy' due to loss of basal tapering and presented a ~30% wider basal diameter of the mature IMC due to lack of BC constriction (Fig. 8f). In conclusion, BCC1 is required for either MyoJ and Cen2 recruitment to

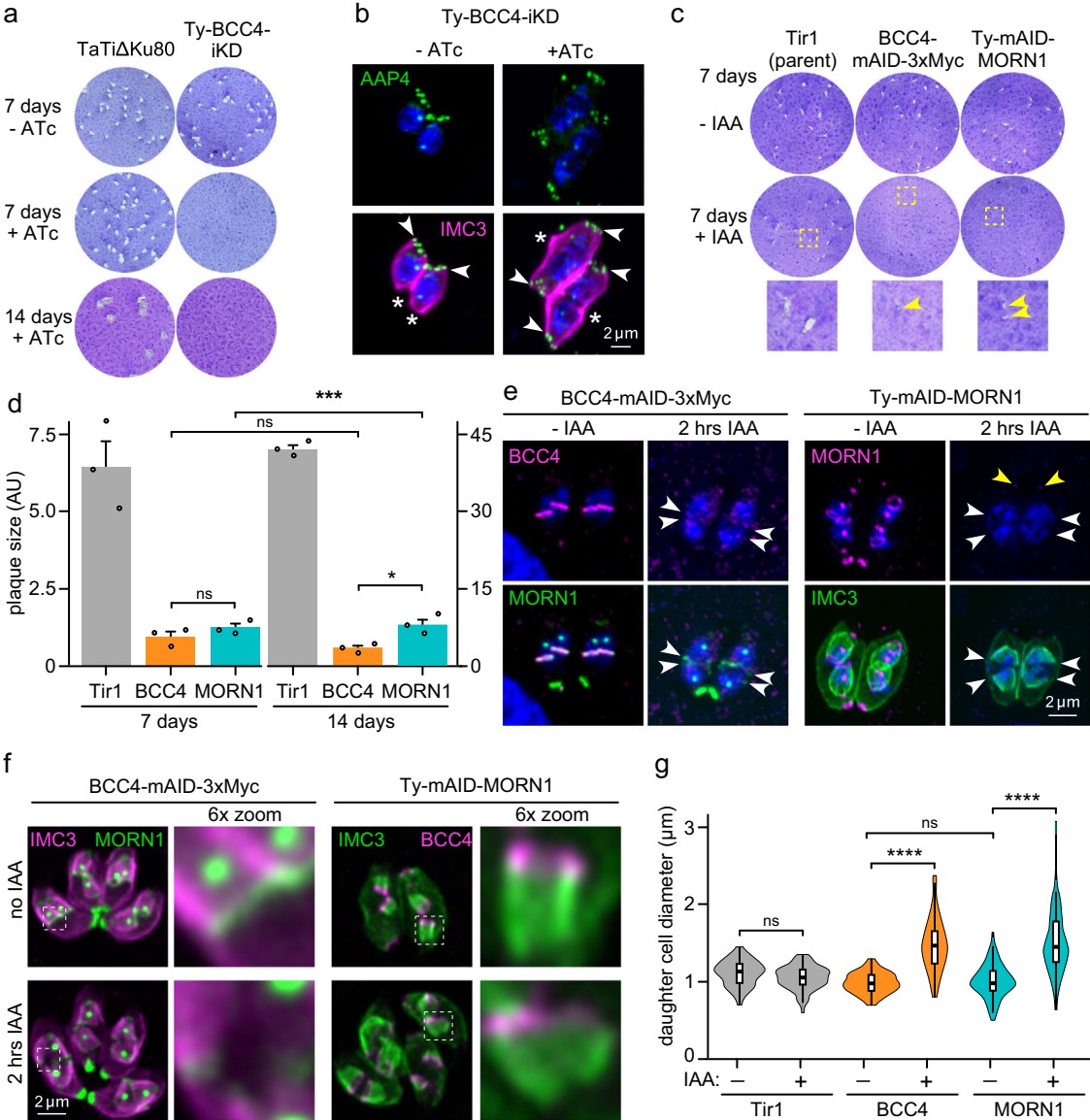

**Fig. 6 | BCC4 is essential for completing the lytic cycle by way of BC constriction. a** Replacement of the BCC4 promoter with a TetO7sag4 ATc-regulatable promoter (and simultaneous N-terminal Ty-tag insertion) completely abolishes plaque forming capacity. Representative example of $n = 3$ biological replicates shown. **b** 24 hrs BCC4 knock-down results in double-headed daughter formation. AAP4 clusters mark the apical end (arrowheads); basal end is marked with asterisks. DNA is stained with DAPI (blue) and applies to all blue stain panels. **c** Plaque assays of parasites wherein either BCC4 or MORN1 is fused with the mini auxin inducible degron (mAID) present small plaques (yellow arrowheads in magnified boxes) in both lines after 7 days + IAA. Representative example of $n = 3$ biological replicates is shown. **d** Plaque sizes after 7 and 14 days + IAA highlight that BCC4-mAID stops proliferating, whereas MORN1-mAID continuous slow proliferation. Left- and right-axis: 7- and 14-day scales, respectively. Tir1 (parent) represents wild-type control. $n = 3$ biological replicates. Data are presented as mean values +SEM. Statistical significance tested by one-way ANOVA ($F_{11,24} = 461.8$, $p = 2^{-16}$) and post hoc Tukey's test. ns = not significant, *$p$-value = 0.0328 (BCC4$_{14\text{days}}$ vs MORN1$_{14\text{days}}$),

***$p$-value = 0.0006 (MORN1$_{7\text{days}}$ vs MORN1$_{14\text{days}}$). **e** Fast kinetics of mAID mediated knock-down of BCC4 and MORN1 for 2 hrs. BCC4 vanished from the BC (YFP-MORN1: arrowheads). Ty-mAID-MORN1 depletes from the centrocone and BC (white arrowheads) but not from the apical end (yellow arrowheads). **f** BCC4 and MORN1 mAID parasites reciprocally endogenously expressing YFP-MORN1 or 3xMyc-BCC4 were 2 hrs IAA treated and co-stained with IMC3 antiserum. Both lines display wider daughter basal ends. Dotted line boxes mark representative basal daughter ends in the left panels, and correspond with zoom panels. **g** Violin plots of quantified daughter BC opening defects shown in panel '**f**. For the box plots, the whisker bar indicates median; whisker box upper and lower box corner represent 75th and 25th percentile, respectively. Whiskers mark highest and lowest value measured. Significance tested by one-way ANOVA ($F_{5,426} = 60.44$, $p = 2^{-16}$) and post hoc Tukey's test. ns = not significant, ****$p$-value <0.0001. $n$ = number of daughter cell basal ends measured. Tir1-IAA ($n = 76$), Tir1+IAA ($n = 77$), BCC4-IAA ($n = 74$), BCC4 + IAA ($n = 93$), MORN1-IAA ($n = 65$), MORN1 + IAA ($n = 48$). Source data are provided as a Source data file.

the BC or maintaining stability of this complex. In either scenario BCC1 is required for final BC constriction.

## Discussion

Proximity biotinylation paired with experimental validation provided a highly refined architectural map of the BC that we used as a road map for functional dissection. This uncovered the BC's essential role in cell division and revealed new cell division phenomena, summarized in Fig. 9. Regarding architecture of the BC, we defined four different BCSCs that fit with the experimental co-localization experiments of defining BCSC components (Fig. 1a)[7]. We can assign the most critical function to BCSC-1 containing MORN1 and BCC4, as they are essential to maintain BC integrity needed to successfully complete cell division. There are no apparent specific or dedicated functions organized in

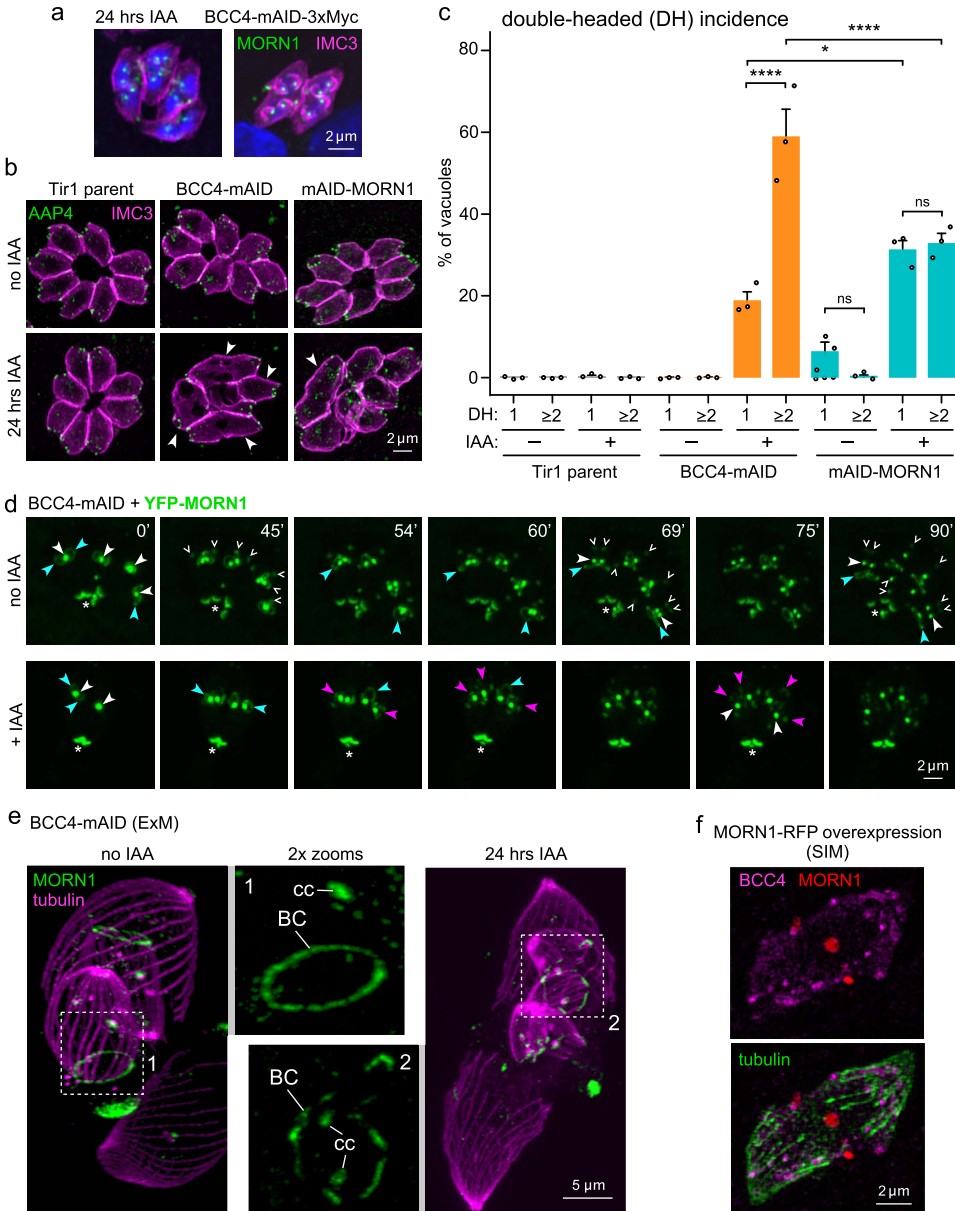

**Fig. 7 | BCC4 and MORN1 are both needed to maintain BC integrity during endodyogeny. a** BCC4-mAID parasites co-expressing YFP-MORN1 were 24 hrs IAA treated and co-stained with IMC3 antiserum. Two different multi-head phenotype development stages are shown. Note the bell-shape daughter bud appearance due to loss of basal integrity. DNA is stained with DAPI (blue). **b** BCC4-mAID and Ty-mAID-MORN1 parasites IAA treated for 24 hrs exhibit different double-headed phenotype (arrowheads) frequencies. AAP4 marks the apical ends. $n = 3$ biological replicates. **c** Double-headed quantification in BCC4 and MORN1 parasites 24 hrs IAA treated stained with IMC3 and AAP4 antisera. 100 random vacuoles were quantified for zero, one (1 DH), and multiple (≥2DH) double-headed parasites. Average of $n = 3$ biological replicates + SEM. Significance tested with one-way ANOVA ($F_{11,24} = 69.68$, $p = 1.85^{-15}$) with post hoc Tukey test. ns = not significant, $*p = 0.0347$ (BCC4$_{1DH}$ vs MORN1$_{1DH}$), $****p < 0.0001$. Source data are provided as a Source data file. **d** Select time-lapse microscopy (Supplementary Movies 2 and 3) panels of BCC4mAID-

3xMyc parasites co-expressing YFP- MORN1. IAA was added at $t = -2$ h. MORN1 localization is annotated in select panels by open arrowheads (budding daughter apical end), closed white arrowhead (centrocone), blue arrowhead (budding daughter BC), or magenta arrowhead (fragmented daughter BC). $N = 2$ biological replicates. **e** Expansion microscopy (ExM) of BCC4-mAID-3xMyc parasites expressing YFP-MORN1 co-stained with α-tubulin antiserum (24 hrs ± IAA). MORN1 still localizes to the BC of daughters but ring integrity is lost after BCC4 degradation. Magnified panels 1 and 2 correspond with the boxed areas in the left and right panel. BC, basal complex; cc, centrocone (spindle pole). 3D rendered rotations provided in Supplementary Movies 4 and 5. **f** SR-SIM microscopy of parasites harboring endogenously tagged BCC4-3xMyc and transiently (24 hrs) over-expressing exogenous MORN1-mCherryRFP[8] were co-stained with Myc and β-tubulin antiserum. BCC4 does not accumulate with MORN1-mCherryRFP in the arrested BC but scatters along the sub-pellicular microtubules.

---

BCSC-2 or -3 as they do not contain any BC localizing proteins with fitness scores hinting at essential roles, other than that they are possibly critical to interface the BC with IMC cytoskeleton (Fig. 2, Supplementary Fig. 3). BCSC-4 harbors three key proteins (BCC1/MyoJ/Cen2) needed to taper the parasites in the finalizing steps of cell division.

Temporal resolution was gained from tagging genes and tracking their localization throughout daughter development. The collective insights are presented in Supplementary Fig. 6, which highlights four distinct protein recruitment steps to the BC coinciding with the functional steps in the BC: initiation, extension, constriction, and maturation. Although initiation exclusively contains BCSC-1 proteins, the other three steps sample from all four BCSCs, supporting a parallel rather than a sequential BC assembly model. Although the majority of proteins remains associated with the BC upon maturation, several

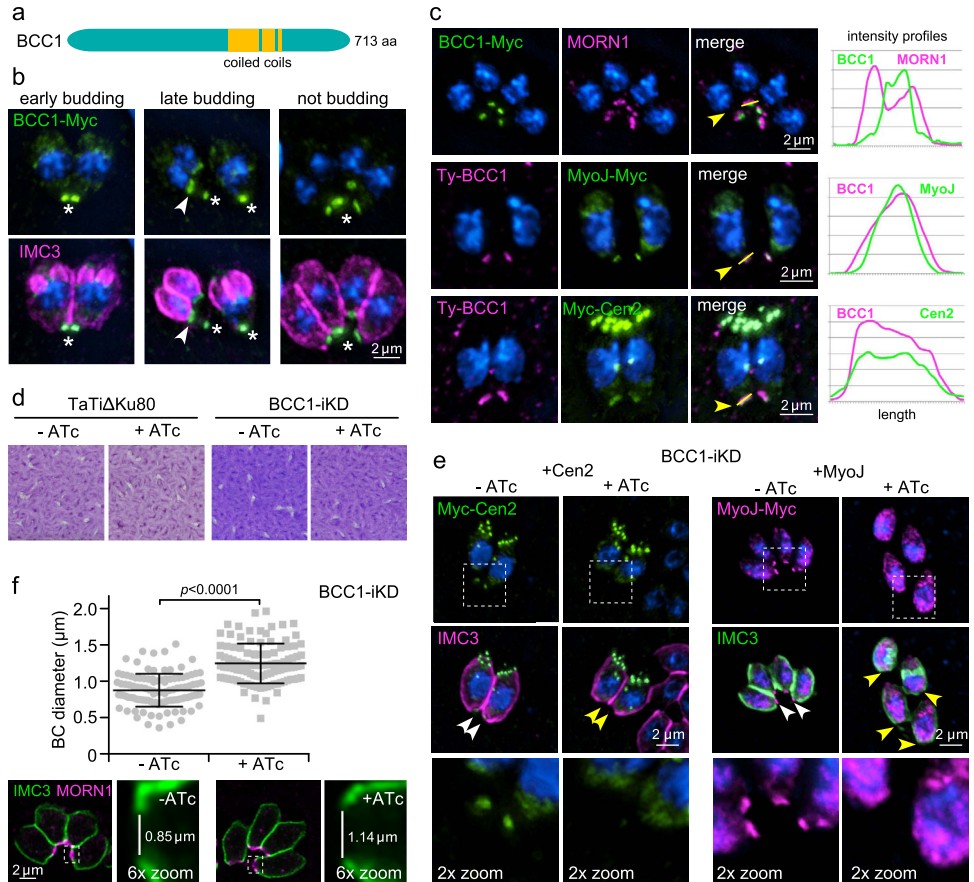

**Fig. 8 | BCC1 acts in the final constriction phase of the BC. a** Schematic representation of BCC1, which only harbors coiled-coils as recognizable domains. **b** BCC1-3xMyc associates with the BC in the second half of division, when daughter cells begin to taper. Asterisks mark BCC1 in the BC of the mature cytoskeleton; arrowheads mark BCC1 in the BC of the budding daughter cytoskeletons. DNA is stained with DAPI (blue) and applies to all panels with blue stain. **c** Co-localization of BCC1-3xMyc with Ty-MORN1 or TetO7sag4-TyBCC1 with MyoJ-3xMyc or 2xMyc-Cen2 demonstrated that BCC1 resides in the most basal BC compartment defined by MyoJ and Cen2. Intensity profiles depict fluorescence intensity (y-axis) over length (yellow bar in adjacent image, marked by yellow arrowhead) on the x-axis. **d** Plaque assays of BCC1 knock-down by tetracycline-regulatable promoter (BCC1-iKD) show no significant reduction in viability over 7 days. $N = 3$ biological replicates. **e** BCC1 knockdown results in the loss of Cen2 and MyoJ from the mature BC. Boxed regions in top panels are magnified in the lower panels. White arrowheads mark Cen2 or MyoJ in the basal complex; yellow arrowheads mark their absence. **f** BCC1 knockdown leads to an impaired constriction of the BC in the conclusion of cell division. BC diameter was measured as illustrated in the bottom panels. Boxed regions in left panels are magnified in the right panels. At least 100 mature BCs were measured and are plotted. Horizontal line marks the average; error bars represent SEM. BCC1-ATc ($n = 108$), BCC1 + ATc ($n = 124$), two-tailed $p$-value < 0.0001. Source data are provided as a Source data file.

proteins are released arguing for roles dedicated to cell division. Specific function for the proteins recruited to the BC in the mature cytoskeleton have been largely elusive[18].

Daughter budding initiates by recruitment of scaffolding proteins to the outer core of the duplicated centrosomes[3]. Here, BCC0 and BCC3 present a distinct 5-fold symmetry. Localization data of other early bud proteins, F-BOX ubiquitin ligase (FBXO1)[40] and two unique apicomplexan proteins, apical cap protein 9 (AC9)[39,41] and IMC32[42], reported a similar spotty appearance at low resolution; using super-resolution we conclusively resolved this as a 5-fold organization. IMC32 and FBXO1 are required for IMC membrane skeleton formation, where AC9 is critical for conoid and sub-pellicular microtubule formation. Here we show that BCC0 and BCC3 define the position of the sutures, and thus are pivotal for defining the alveolar vesicle architecture. BCC0 was prominent in our BC-BioID data but IMC32, AC9, and FBXO1 were absent, which strongly supports our interpretation that BCC0 is foundational for the BC.

One of the most striking observations is that bud growth is bidirectional from the BCC0/3 foundation: ISP1 representing the apical cap assembles and extends in the apical direction (Figs. 3c, 9, panels 4 and 5), whereas the BC marked by MORN1, the longitudinal sutures

marked by both BCC0 and 3 as well as the more basal alveoli marked by IMC3 extend in the basal direction (Figs. 3c, d, 4b–d, 9, panels 4 and 5). This is counter to the current model where extension is considered to occur exclusively in the basal direction[4,5]. However, our model fits with reported observations that did not align with the old model: upon ISP2 depletion, ISP1 still assembles in rings around the centrosome, but budding does not proceed any further[33]. Here, ISP1 gets stuck on the 5-fold symmetrical scaffold, but ISP2 is needed to extend the cytoskeleton. Furthermore, the role of the orthologous ISP proteins in *Plasmodium* ookinete formation, which buds outward from the plasma membrane, has provided an outward push model driven by protein palmitoylation[43]. We think this outward push principle also applies in *T. gondii*, but exclusively to the apical cap. Further support is that both the apical cap and the ookinete membrane skeleton are composed of only a single alveolar vesicle. The more complex alveolar quilt seen in *T. gondii* seems to rely on a different assembly principle, notably the addition of components from the basal end of the bud[44]. Furthermore, DHHC14 in BCSC-2 (Supplementary Fig. 3) is critical to daughter cytoskeleton assembly[26] and fits the new model.

The other progressive insight is that the BC's most critical function is keeping the basal end of the daughter bud together. This

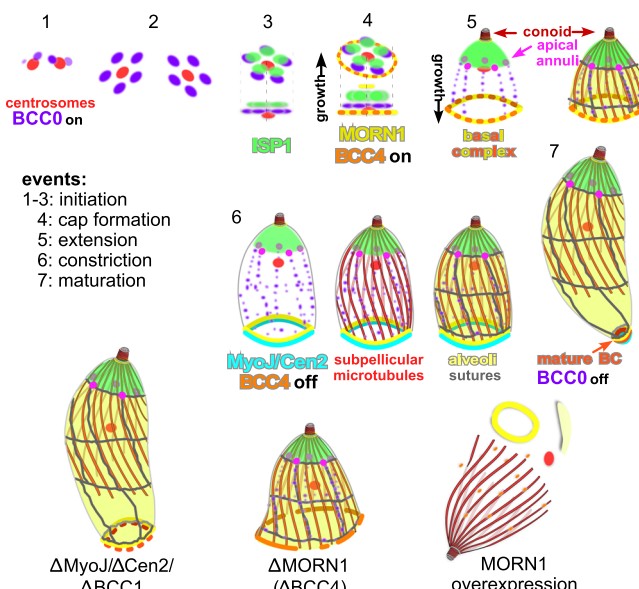

**Fig. 9 | Summarizing schematic.** Only defining components in BC formation and daughter budding are depicted. Note that 'cap extension' (step 4) and 'extension' (step 5) occur simultaneously but progress in opposite directions. The appearance of the name of components coincides with their recruitment to the BC, whereas for BC components that are dynamic, their named recruitment is defined by 'on' and their release by 'off'. The schematics representing the phenotype mechanisms of various deletion mutants, or overexpression (MORN1) are interpreted from the experimental data.

becomes essential at the midpoint of budding when many different proteins are recruited to the BC (Supplementary Fig. 6), including the BCC1/MyoJ/Cen2 complex. We propose that the BC is critical to keep the (+)-ends of the microtubules together, since upon functional BC disruption, the sub-pellicular microtubules appear like an opened umbrella (Fig. 7f)[8]. This would imply that during the first half of division the BC is permitted to expand like a rubber band, but that the band reaches its maximal stretch at the midpoint of cell division, upon which it needs to be reinforced with additional proteins. This explains the fragmentation of the BC upon BCC4 or MORN1 depletion (Fig. 7a–e). MORN1 dimers attain either an extended or a V-shape conformation, which provides a mechanism for a stretchable BC[45]. Another implication of the BC-stretch model is that a motor protein is not needed, which matches our data.

Studies on the BC in *P. falciparum* have identified a set of proteins that is largely not conserved in *T. gondii*[46]. To obtain a bird's eye view of BCC conservation we searched EuPathDB[47] to map conservation patterns. Interestingly, BCCs partitioned into three distinct groups (Supplementary Fig. 7): group 1 is widely conserved across Apicomplexa, group 2 is conserved across all Coccidia, whereas group 3 is narrowly conserved in tissue cyst forming Coccidia dividing by internal budding. This analysis might be skewed by the high incidence of low complexity and coiled-coil regions in the BCCs, and the relatively low number of introns (Fig. 1e) in these proteins indicates they are likely fast evolving. Alternatively, lack of pan-apicomplexan conservation could represent the mechanistic differences between internal and external budding modes[48].

## Methods

### Parasites and mammalian cell lines

Transgenic derivatives of the RH strain were maintained and assessed in hTERT immortalized human foreskin fibroblasts (HFF) except for IFA assays, which were performed in primary HFF cells, largely as previously described[49]. Stable parasite transfectants were selected under 1 μM pyrimethamine, 20 μM chloramphenicol, or a combination of 25 mg/ml mycophenolic acid and 50 mg/ml xanthine (MPA/X).

### Plasmids and parasite strain generation

For endogenous 5′-end tagging of BC genes with BioID2 we used a previously reported method[20]. In short, expression of the selection marker was linked to the integration into a specific gene locus (selection-linked integration (SLI)[50]). A PCR amplicon consisting of the HXGPRT ORF linked to the sequence of the ty-1-bioid2 ORF by the T2A skip peptide sequence was transfected together with a CRISPR/Cas9 plasmid that generated a specific DNA double-strand break around the ATG of the respective gene locus. The PCR amplicon carries 35 bp homologous flanks on each site to facilitate homologous repair. Transfected parasites were selected using MPA/X for expression of HXGPRT under the respective endogenous promoter.

Prey genes were analyzed by endogenous tagging via 3′-end replacement (all oligonucleotides listed in Supplementary Data 2). Homologous 3′end flanks of a given prey gene were cloned via PmeI and AvrII into the integration plasmid to generate 3xMyc or YFP-tagged alleles. 50 μg of plasmid DNA was linearized with a restriction enzyme digest before transfection in RHΔKu80 parasites.

Conditional knockdown parasite lines were generated by PCR-amplifying the DHFR-tetO7-sag4 sequence from a donor plasmid including 35 bp flanks for homologous repair. A CRISPR/Cas9 plasmid co-transfected with the amplicon generated a specific double-strand break around the ATG of the respective gene and allowed for promoter swap in TATiΔKu80 parasites[51].

Endogenous 3′end tagging with mAID were done as follows. The DNA sequence of the minimal Auxin inducible degron (mAID)[52] was amplified from a donor plasmid (kind gift of Dr. Lourido, Whitehead Institute) and cloned in frame with the PmeI/AvrII restriction sites of the 3xMyc-DHFR 3′end integration plasmid. The homologous 3′flank used for endogenous tagging of TGGT1_229260 with 3xMyc was cloned into the newly established plasmid and linearized as previously mentioned. 50 μg of plasmid DNA was used for transfection of Tir1ΔKu80 parasites[38] and transfected cell lines were selected with Pyrimethamine. To induce specific protein degradation, parasites were incubated with 500 μM IAA (in 100% ethanol) for times indicated in the result section. To endogenously tag MORN1 or BCC4 in RH Tir1-3xFLAG parasites, we generated a plasmid that linked the HXGPRT or DHFR/TS ORF via the T2A skip peptide to the YFP ORF, the 3xMyc or 5xV5 epitope tag sequence. A PCR amplicon was transfected together with 40 μg of a CRISPR/Cas9 plasmid that generated a DNA double-strand break around the ATG of either MORN1 or BCC4. Parasites expressing YFP/5xV5-MORN1 or 3xMyc-BCC4 were selected with MPA/Xan or Pyr as mentioned above.

### BioID sample preparation and mass spectrum analysis

Biotin labeling was done in two biological replicates (+biotin condition) and one biological replicate (-biotin condition) as reported before[20]. Each biological replicate was run as two technical replicates on the mass spectrometer. Biotinylated proteins were identified as previously reported[20]. In short, parasites expressing BioID2-fusion proteins were grown overnight ±150 μM biotin and extracellular parasites were harvested, filtered through a 3 μm membrane, and mechanically lysed in 1% SDS in resuspension buffer (150 mM NaCl, 50 mM Tris-HCl pH 7.4). 1.5 mg of total protein lysate was used for streptavidin pull-down with Streptavidin-agarose beads (Fisher). Proteins bound on beads were reduced, alkylated, and digested with Trypsin. Digested peptides were used for LC-MS/MS analysis, which was performed on an LTQ-Orbitrap Discovery mass spectrometer (ThermoFisher) coupled to an Agilent 1200 series HPLC. Samples were pressure loaded onto a desalting column (250 μm fused silica), packed with 4 cm of Aqua C18 reverse phase resin (Phenomenex). Peptides were then eluted onto a 100 μm fused silica, biphasic column (5 μm tip, packed with 10 cm Aqua C18

resin and 4 cm Partisphere strong cation exchange resin [Whatman] using a gradient of 5–100% Buffer B in Buffer A (Buffer A: 95% water, 5% acetonitrile, and 0.1% formic acid; Buffer B: 20% water, 80% acetonitrile, and 0.1% formic acid). Peptides were finally eluted from the strong cation exchange resin onto the Aqua C18 resin and into the mass spectrometer using four salt steps (95% water, 5% acetonitrile, 0.1% formic acid, and 500 mM ammonium acetate). The flow rate through the column was set to -0.25 µl/min, and the spray voltage was set to 2.75 kV. With dynamic exclusion enabled, one full MS scan (FTMS; 400–1800 MW with a resolution of 30,000) was followed by seven data-dependent scans (ITMS) of the $n^{th}$ most abundant ions. Generated MS RAW files were converted to MS2 files using RawConverter[53]. The tandem MS data were searched using the SEQUEST algorithm[54] using a concatenated target/decoy variant of the *Toxoplasma gondii* GT1 ToxoDB-V29 database (ToxoDB.org). The SEQUEST search was done with the following settings; a static modification of +57.02146 on cysteine was specified to account for alkylation by iodoacetamide, the precursor and fragment ion mass tolerance was set to 50 ppm, up to one missed cleavage was allowed, and no specific minimum peptide length was defined for the peptide identification. SEQUEST output files were filtered using DTASelect 2.0[55]. Reported peptides were required to be unique to the assigned protein (minimum of two unique peptides per protein) and discriminant analyses were performed to achieve a peptide false-positive rate below 5%. Data for Ty-BioID2-Cen2 and YFP-BioID2 were used from a previous study[20].

## Analysis of mass spectrometry data by probabilistic calculation of interactions

Spectral counts of unique proteins (Supplementary Data 3) were used to determine probability of interaction for given bait and preys using SAINTexpress[22]. SAINTexpress was executed using the −L4 argument, compressing the four largest quantitative control values of a given prey in one virtual control. The resulting SAINTexpress matrix (Supplementary Data 4) was visualized using the Prohits-viz online suite[24]. The following preys were manually deleted from the analysis using the Zoom function in Prohits-viz: HXGPRT, TGGT1_269600 (annotated as biotin enzyme), TGGT1_289760 (annotated as biotin-synthase). The dot plot and the correlation map in Supplementary Fig. 3 were generated by including a cytosolic BioID2-YFP control[20] in the SAINT analysis. Preys were manually arranged into BC subcomplexes for Fig. 2b, the entire resulting dot plot is shown in Supplementary Fig. 3b.

## (Immuno-) fluorescence microscopy

Indirect immunofluorescence assays were performed on intracellular parasites grown overnight in 6-well plate containing coverslips confluent with HFF cells fixed with 100% methanol (unless stated otherwise) using the following primary antisera: mouse α-Ty clone BB2 (1:500; kindly provided by Dr. Lourido, Whitehead Institute), MAb 9E10 α-cMyc (1:50; Santa Cruz Biotechnology), MAb 9B11 α-cMyc Alexa488 (A488) conjugated (1:100; Cell Signaling Technologies), mouse α-V5 clone SV5-Pk1 (1:500, BioRad), rabbit α-β-tubulin (1:1000; kindly provided by Naomi Morrissette, University of California, Irvine[56]), rat α-IMC3 (1:2000[7]), rabbit α-IMC3 (1:2000; generated against the N-terminal 120 amino acids fused to His6, generated as described[7]), rabbit α-human-Centrin2 (1:1000; kindly provided by Iain Cheeseman, Whitehead Institute), and guinea pig α-AAP4 (1:200)[20]. Streptavidin-A594 (1:1000; ThermoFisher), A488, A594, or A633 conjugated goat α-mouse, α-rabbit, α-rat, or α-guinea pig were used as secondary antibodies (1:500; Invitrogen). DNA was stained with 4′,6-diamidino-2-phenylindole (DAPI). A Zeiss Axiovert 200 M wide-field fluorescence microscope was used to collect images, which were deconvolved and adjusted for phase contrast using Volocity software (Improvision/Perkin Elmer). SR-SIM or Zeiss Airyscan imaging was performed on intracellular parasites fixed with 4% PFA in PBS and permeabilized with 0.25% TX-100 or fixed with 100% methanol. Images

were acquired with a Zeiss LSM880 with ELYRA S.1 (SR-SIM) and Airyscan system in the Boston College Imaging Core in consultation with Bret Judson. All images were acquired, analyzed, and adjusted using ZEN software and standard settings. Final image analyses were made with FIJI software.

## Live cell microscopy

Live cell microscopy of RH Tir1-3xFLAG BCC4-mAID-3xMyc parasites expressing YFP-MORN1 was done using a Zeiss LSM880 with Airyscan unit in the Boston College Imaging Core in consultation with Bret Judson. Parasite dynamics were recorded using the "Airyscan fast" settings, in an incubation chamber set to 37 °C. Parasites were grown overnight under standard culture conditions in 3 ml live cell dishes (MatTek). On the next day, culture medium was replaced with live cell imaging medium (DMEM without phenol red, 20 mM HEPES pH 7.4, 1% FBS, Penicillin/Streptomycin, and Fungizone). To induce protein degradation, parasites were treated with 500 µM IAA (in 100% ethanol) 2 hrs before imaging started. The resulting data were deconvolved using standard Airyscan settings and movies processed with FIJI software.

## Quantification of daughter basal end diameter

BCC4-mAID-3xMyc parasites endogenously expressing YFP-MORN1, Ty-mAID-MORN1 parasites endogenously expressing 3xMyc-BCC4 or Tir1 parental parasites were seeded on coverslips with HFF host cells and grown overnight. At the next day, protein degradation was initiated with 500 µM IAA and parasites were incubated for two more hours. Cells were fixed with 4% PFA and stained with anti-IMC3 and anti-Myc (in case of Ty-mAID-MORN1 parasites) serum and images of dividing parasites were acquired using the Zeiss LSM880 Airyscan. Images were analyzed in FIJI[57] and basal diameter measured. Resulting data were visualized with the ggplot package in R[58].

## Expansion microscopy

Expansion of *Toxoplasma* tachyzoites was achieved by following recently published protocols[39,59,60]. Briefly, tachyzoites growing in HFF cells for 24 hrs ±IAA were released by syringe lysis and filtered through a 12 µm membrane. Free tachyzoites were allowed to settle on poly-L-lysine coated coverslips at 4 °C for 30 min, followed by fixation in −20 °C methanol for 7 min. The U-ExM protocol was started by incubating coverslips in 2x solution (2% formaldehyde, 1.4% acrylamide (AA) in PBS) for 5 hrs at 37 °C. Gelation was done in Monomere solution (19% (w/w) sodium acrylate, 10% (w/w) AA and 0.1% (w/w) BIS-AA in PBS) complemented with ammonium persulfate (APS) and tetramethylethylenediamine (TEMED) for 1 hr at 37 °C, followed by incubation in denaturation buffer (200 mM SDS, 200 mM NaCl, 50 mM Tris, pH 9) at 95 °C for 90 min. Gels were incubated for a first round of expansion in ddH$_2$O overnight and washed twice in PBS on the next day. As a primary antibody, mouse α-alpha-tubulin (12G10, 1:250) and rabbit α-GFP (Torrey Pines) were used to stain parasite microtubules and YFP-MORN1, respectively. Gels were incubated in 2% BSA in PBS with primary antibodies at 37 °C for 3 hrs, washed three times with PBST (1xPBS + 0.1% Tween20), and incubated for 3 hrs at 37 °C in 2% BSA in PBS complemented with secondary antibody (goat anti-rabbit-Oregon Green, goat anti-mouse-A594, Invitrogen). Gels were washed three times in PBST before a second overnight expansion in ddH$_2$O. For imaging, gels were mounted in 35 mm glass bottom microwell dishes (MatTek) and imaged on a Zeiss LSM880 with Airyscan unit using standard settings for image acquisition and Airyscan deconvolution. All imaging was done in the Boston College Imaging Core with help/advice of Dr. Bret Judson.

## Statistics and reproducibility

Graphical data were generated with GraphPad Prism v6.0h software or R statistical software v3.6.3 with packages ggplot2[58] and rstatix. Tables

were visualized with Microsoft Excel. Statistical significance was determined by unpaired, two-tailed *t*-test with Welch-correction or one-way ANOVA with post hoc Tukey's HSD test, in case three or more conditions were compared. *P*-values <0.05 were considered statistical significant. Unless otherwise stated, three biological replicates ($n=3$) were used for the experiments and data presented as mean +/− SEM. All microscopy images are representatives of at least two independent experiments. All experiments resulted in comparable results.

## Reporting summary

Further information on research design is available in the Nature Research Reporting Summary linked to this article.

## Data availability

The mass spectrometry proteomics data have been deposited to the ProteomeXchange Consortium via the PRIDE[61] partner repository with the dataset identifier PXD031116 *T. gondii* genome information can be found in ToxoDB (https://toxodb.org). Source data are provided with this paper.

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

## Acknowledgements

We thank Sudeshna Saha, Ciara Bauwens, Eleni Kisty, and Adam Mehal for technical support, Bret Judson and the Boston College Imaging Core for infrastructure, support and fruitful discussions, Drs. Cheeseman, Lourido, and Morrissette for sharing reagents. This study was supported by grants of the Deutsche Forschungsgemeinschaft, the American Heart Association (post-doctoral fellowship 17POST33670577), and the Knights Templar Eye Foundation (early career starter grant) to K.E.; and by a National Science Foundation (NSF) Major Research Instrumentation grant 1626072, National Institute of Health grants AI110690, AI128136, AI144856, and AI152387 to M.J.G. The funders had no role in study design, data collection and analysis, decision to publish, or preparation of the manuscript.

## Author contributions

K.E. and M.J.G. conceived the approach, analyzed and interpreted the data, and co-wrote the manuscript. K.E. designed and generated all *T. gondii* mutants and data. T.B. and E.W. performed and analyzed the mass spectrometry data. C.M. assisted with data analysis, presentation and assessed the conservation of the BCCs. All authors proofread the manuscript.

## Competing interests

The authors declare no competing interests.
