## [Peer Review File · Nature Communications]

REVIEWER COMMENTS

Reviewer #1 (Remarks to the Author):

Review of Engelberg et al, NCOMMS-21-50887

The manuscript provides a detailed evaluation of the *Toxoplasma gondii* basal complex using proximity labeling with follow up genetic studies. The authors find four different subcomplexes that differ in composition, timing, and location during basal complex function. Multiple new basal complex proteins (BCC0-11) are identified, and the location and function of three newly identified proteins, BCC0, BCC1, and BCC4, are investigated. This leads the authors to propose a “rubber band” model for BC function. In addition to identifying and characterizing multiple new components, the finding that the budding daughters form bidirectionally is really interesting. The investigators also provide strong data for differential functions of the BC sub-complexes.

Major comments:

1. The localization of MORN1 should be shown in the BCC0-iKD parasites to see if BCC4 knockdown prevents the correct localization of MORN1.
2. The main findings in the manuscript are present in the microscope images. These images can be difficult to interpret for a non-aficionado. It would be helpful to have a schematic of the structures for each figure, and ideally for column (and/or) of each panel. This would tremendously make the data more approachable.
3. BCC3 appears not to colocalize that strongly with MORN1. It looks more like an IMC protein.
4. What is the implication of the phenotype of “stumpy” daughters for BCC1 loss? There is not a major fitness loss for this protein. Is the implication that the size of the daughter cell basal end is not important?

Minor comments:

1. Fig 1a: the zoomed images of the basal complex markers should include a scale bar. Ideally, these images should all be presented at the same zoom.
2. Fig 1b: while Ty-BioID2-Cen2 has been previously reported, it would be nice for completeness to show an image of the correct localization in this figure as well.
3. Fig 2a: “loser cluster” should be “looser cluster”

4. Fig S1: It would look nicer, perhaps, if BCC11 were moved to the bottom of the right column. It currently seems out of order.
5. Line 211: this should refer to 3e not 3d for AAP4.
6. Line 216: this is somewhat more definitive than the data allow. It would be better to say the “likely” or “potentially set to form” as opposed to “set to form”.
7. Fig 4b the second row in the “late” and “maturing” columns need the IMC3 staining shown. The formatting of these panels seems incorrect.
8. Fig 6a – this result should be repeated, not n=1 experiment. Fig 6c should say that it is a representative example of the n=3 biological replicates.
9. Fig 7a (and 7b) – why is the “24” in white font/black background? Also, the parental parasite looks like it is Tirl and Tir1.
10. Line 328 – It would be better to say “An essential role of the BC” rather than “The essential role”. As stated currently, the sentence implies that the sole essential role of the BC is to bundle the microtubules.

Reviewer #2 (Remarks to the Author):

The manuscript by Engelberg et al extends our knowledge of the composition of the basal complex (BC) in *Toxoplasma* by identifying additional components and their interactions during development. The BC is a unique organelle at the posterior end of the mature parasite and proteins localizing there undergo dramatic reorganization during cell division. The process of daughter formations in *Toxoplasma* is distinct from the typical actin-furrow mechanism of cell division used by higher eukaryotes and therefore is potentially of interest. Unfortunately, the findings are largely descriptive and they do not provide mechanistic insight into any of the components. This work will likely be highly valuable for a select field of specialist interested in this process, but it does not establish a new biological paradigm nor does it explain how the components work to promote cell division.

Major concerns:

- 1 The MS data appear to be thoroughly conducted, but I am concerned about the relatively small number of biological replicates (2) that would appear to limit statistical analysis. Although the authors report using FDR values in SAINT to analyze the data, it would appear they had to combined

technical and biological replicates to do so. This practice is likely to inflate the significance estimates and is not reliable.

2) A number of new components of the BC are identified, and they contain interesting domains that would predict various biochemical functions (kinases, phosphatases, calcium binding motifs). None of these are followed up in any meaningful way. Instead the authors focus on genetic knock downs to test essentiality (which is largely predictable from the fitness scores) and to examine altered patterns of cell division.

3) Much of the logic and analyses follows what has been previously reported by this group and others for proteins such as MORN. In fact, one of the new components (BCC4) phenocopies MORN- although this does not tell us much about how it functions. The resulting summary is a highly descriptive interaction map of some of the players involved in BC formation and cell division. They do little to explain how the process actually works.

Reviewer #3 (Remarks to the Author):

The authors conduct a proximity labeling analysis of the basal complex (BC) of *Toxoplasma Gondii* and identify multiple novel subunits present in this structure. Overall this paper is well-written and provides useful data to the community. Specific concerns below:

1. The authors are over-reliant on the use of spectral counts for quantitation. In particular, identifying candidates in Fig. 1A based on NormSpC and not more rigorous statistical analyses of enrichment is concerning. Considering the authors use SAINT later in the manuscript, it is not clear why they did not use it in Fig 1 also instead of arbitrarily focusing on top 25 hits.

2. It should be noted that although the use of delta spectral counts as a metric for differential abundance is crude at best and better done using more robust statistical frameworks (like SAINT or Compass), the biochemical validation of the results in this study are convincing which greatly lessens concern over the analysis of the proteomic data.

Point by point rebuttal of Engelberg et al, NCOMMS-21-50887A

Reviewer #1:

Major comments:

1. The localization of MORN1 should be shown in the BCC0-iKD parasites to see if BCC4 knockdown prevents the correct localization of MORN1.

We agree that this is an important experiment, and have added this as a new panel (Fig 3i). In addition, we have added annotation in Fig 3h, pointing out maternal and daughter apical annuli (stained with AAP4). Collectively, the AAP4, MORN1, and IMC3 stains show that the daughter IMC initially forms, but that the apical annuli and MORN1 in the daughter buds are mis-organized, which when budding progresses, results in a misshapen daughter bud IMC scaffolds. This is already somewhat visible after 24 hrs of BCC0 knock-down when buds are wider (panels 3g, h, i) but increases dramatically after 48 hrs (panel 3j). As such, BCC0 is critical for correct assembly of the BC and the annuli, which in turn affects the stability of the IMC skeleton. We have clarified this interpretation in the revised manuscript.

2. The main findings in the manuscript are present in the microscope images. These images can be difficult to interpret for a non-aficionado. It would be helpful to have a schematic of the structures for each figure, and ideally for column (and/or) of each panel. This would tremendously make the data more approachable.

*We can appreciate this comment and do agree that especially when there is no parasite outline (e.g. marked by a cortical cytoskeleton marker such as IMC3) it is hard to orient what is going on. As such, we have added schematics and/or guides as suggested in those panels where a cortical reference marker was missing (Fig 3b, c, d, h, i; Fig 5c, d, e). Specifically in Fig 3b, the first figure in the manuscript going through sequential steps of BC development, is now used to provide a schematic to provide a reference for perspective, together with corresponding 200% zoom panels of the events occurring around the centrosome where the BC forms, and mitosis occurs. Moreover, we now direct the reader more explicitly to Figs. 9 and S5 for an overview of the *T. gondii* cell division process.*

3. BCC3 appears not to colocalize that strongly with MORN1. It looks more like an IMC protein.

We agree that BCC3 for most of the time is on the bud/sutures, but it clearly does localize to the BC in “late budding” and “maturing” daughter parasites in presented in Fig 4b. Our nomenclature and assignment as a BC protein is driven by the way we discovered it, following the (informal) conventions in the field.

4. What is the implication of the phenotype of “stumpy” daughters for BCC1 loss? There is not a major fitness loss for this protein. Is the implication that the size of the daughter cell basal end is not important?

Indeed, BC diameter at late stage of cell division is not preventing the completion of cell division as also seen upon MyoJ or actin depletion (PMID: 28593938, PMID: 28322189). However, preventing BC formation by depletion of MORN1 or BCC4 results in much, much wider basal ends that results in incomplete cell division (double-headed parasites) that is fatal.

The BCC1 phenotype is indeed the same as observed upon MyoJ or actin deletion. As such, it points at a role of BCC1 in either recruiting or stabilizing the MyoJ/Cen2 complex at the BC. We have clarified this interpretation better in the text.

Minor comments:

1. Fig 1a: the zoomed images of the basal complex markers should include a scale bar. Ideally, these images should all be presented at the same zoom.

We have revised this Figure, checked that all sizes are the same, and added scale bars to the zoom images.

2. Fig 1b: while Ty-Biold2-Cen2 has been previously reported, it would be nice for completeness to show an image of the correct localization in this figure as well.

Although we do not disagree with this assessment, in the interest of space we feel not including the same published data is our preferred way to go.

3. Fig 2a: “loser cluster” should be “looser cluster”

Revised as suggested.

4. Fig S1: It would look nicer, perhaps, if BCC11 were moved to the bottom of the right column. It currently seems out of order.

Revised as suggested.

5. Line 211: this should refer to 3e not 3d for AAP4.

Thanks for catching this mis-reference; corrected as suggested.

6. Line 216: this is somewhat more definitive than the data allow. It would be better to say the “likely” or “potentially set to form” as opposed to “set to form”.

We agree with this assessment of the data and have included the word “potentially” as suggested.

7. Fig 4b the second row in the “late” and “maturing” columns need the IMC3 staining shown. The formatting of these panels seems incorrect.

We have revised Fig 4b so that all stages show the same channel combinations, as suggested.

8. Fig 6a – this result should be repeated, not n=1 experiment. Fig 6c should say that it is a representative example of the n=3 biological replicates.

We repeated the experiment of panel 6a two more times and now include the followed description for both panels 6a and 6c: “Representative example of n=3 biological replicates shown.”

9. Fig 7a (and 7b) – why is the “24” in white font/black background? Also, the parental parasite looks like it is Tirl and Tir1.

We used the white font with black background to emphasize the exact time of induction used. Based on this comment we realize this generates unwanted questions, and as such we have removed this emphasis. We also corrected the Tir1.

10. Line 328 – It would be better to say “An essential role of the BC” rather than “The essential role”. As stated currently, the sentence implies that the sole essential role of the BC is to bundle the microtubules.

The reviewer is correct in this observation and we have revised the text accordingly.

Reviewer #2:

Major concerns:

1 The MS data appear to be thoroughly conducted, but I am concerned about the relatively small number of biological replicates (2) that would appear to limit statistical analysis. Although the authors report using FDR values in SAINT to analyze the data, it would appear they had to combined technical and biological replicates to do so. This practice is likely to inflate the significance estimates and is not reliable.

We indeed only did two biological replicates and used the technical replicates in the SAINTexpress analysis, which might have inflated the significance estimates. However, we feel that these concerns are mitigated by the following factors:

- 1. The biological replicates resulted in reproducible data for the most abundant proteins. To this end, we have added mass spectrometry spectral count plots of the two biological replicate experiments for each bait (new Fig S1) to underscore the high replicative agreement between the two biological replicates.*
- 2. Rigor is provided by using a total of 6 BC bait proteins distributed across the different BC sub-complexes, next to a cytoplasmic control.*
- 3. Further rigor and depth is added by the iterative, reciprocal experimental approach we took by using newly discovered BCC1 and BCC2 as baits in a second round of BioID experiments.*
- 4. Our goal was to cast an as wide net as possible to capture putative BC proteins in the SAINTexpress analysis. The major challenge was actually derived from the used of MORN1 and Cen2 as baits, which due to their non-exclusive localization and even overlapping localization early in cell division, provided spurious hits for MORN1 in the centrocone and for Cen2 mostly in the apical annuli (Fig S3; now S4). We tried minimizing these by playing with the analyzes settings by trial and error; in fact, to emphasize this challenge in our data we do include two different analyses in the manuscript, one provided in Fig 2 and another in Fig S2. This trial and error of the settings driven by experimental validations learned that non-BC proteins in the SAINTexpress set were almost exclusively driven by one of the two non-exclusive BC proteins, MORN1 or Cen2 (Fig S2b, S3; now S3b, S4). In particular the dot-blot in Fig*

S2b (now S3b) is strongly illustrating this point. We would like to point out that we actually took advantage of this and serendipitously mapped the apical annuli proteins through Cen2, which was reported in a separate manuscript (PMID 31470470). Therefore, weeding our false BC assignments was the top priority, and in that process, we experimentally validated all putative BC assignments in across several different SAINTexpress settings.

- 5. Finally, of those proteins experimentally validated to localize to the BC, our SAINTexpress based assignments of BC proteins to the BCSC sub-complexes were in all cases tested/validated. We agree this part was not done in exhaustion, which is complicated by the dynamics and kinetics of the BC proteome, however we did not find any BC proteins that did not localize to the BCSC we predicted. We therefore believe our BCSC assignments are robust.*

Overall, we cannot exclude that we might have missed a (minor) number of BC proteins in our analysis. However, our analysis is supported by a very comprehensive set of validation experiments and, in the final version of the manuscript, we only made two tentative assignments solely on the SAINTexpress analysis, one of which we have since experimentally confirmed (not shown). We therefore are very confident our overall analysis is robust and is experimentally validated and an additional biological replicate would only provide very minimal extra rigor.

- 2) A number of new components of the BC are identified, and they contain interesting domains that would predict various biochemical functions (kinases, phosphatases, calcium binding motifs). None of these are followed up in any meaningful way. Instead the authors focus on genetic knock downs to test essentiality (which is largely predictable from the fitness scores) and to examine altered patterns of cell division.*

Our key question was to figure out why the BC is essential in cell division: our starting point was that scaffolding protein MORN1 is essential, but contractile protein MyoJ was not. This did not add up. Once we had a list of BC components, the fitness score was the fastest way to identify essential proteins: of the three severe fitness conferring BC proteins, BCC4 was the only one not studied before (others were MORN1, DHHC14, IMC29 [not a BC protein], ISAP1 [not cell division related], next to a TCP1 chaperone protein). The discovery of BCC4 as one of the few essential proteins in the BC proteome now provides with a mechanism to fill this void.

We completely agree that the proteins with functional domains are worthy of follow up. However, they seem to have redundant functions based on their fitness scores, which would likely not provide strong phenotypes by simple gene expression disruption and would require extensive follow up experiments to determine their role in the BC. Actually, we were really surprised by this: our interpretation is that the role of the BC in cell division is very important and covered by a highly redundant signaling network of non-essential kinases and phosphatases, which will direct our future work.

- 3) Much of the logic and analyses follows what has been previously reported by this group and others for proteins such as MORN. In fact, one of the new components (BCC4) phenocopies MORN- although this does not tell us much about how it functions. The resulting summary is a highly descriptive interaction map of some of the players involved in BC formation and cell division. They do little to explain how the process actually works.*

The major discoveries are that there are surprisingly few essential players in the BC (see reply above), and only 1 that was not studied before (BCC4). Besides this parts list, we do make several mechanistic observations not reported before:

1. We do show that BCC4 together with MORN1 is essential for the 'contractile ring' function of the BC: notably: no motor protein is needed for this, which is a major, and new mechanistic insight.
2. The 5-fold symmetry as seen in the apical annuli and IMC-alveolar vesicle suture patterns as seen in the mature parasite is the first structure that starts forming the daughter cytoskeleton bud. Although hints of this were visible in previous reports, nobody specifically commented on this feature as the pictures were of lower resolution and not defined enough. We increased the imaging power with SR-SIM to provide much more advanced resolution that now permits firm conclusions on how these early steps are organized.
3. Another major new insight is that the cap alveolus buds in the apical direction from the 5 initial foci, whereas the rest of the cytoskeleton assembles in the basal direction from the 5 foci. We hypothesize in the discussion that the single alveolus membrane cytoskeleton seen in Plasmodium parasites seems to mirror what we see for the cap alveolus. In particular this process has been studied in ookinete formation of Plasmodium, and has been shown to be an outward budding force (i.e. apical direction), driven by palmitoyl transferase (PMID: 34467559). This parallels T. gondii in that DHHC12 and DHHC14 palmitoyl transferase have been shown to be critical for daughter budding, but this far no specific role could be assigned to them. The T. gondii bi-directional budding mechanism is completely novel.

Indeed, our work raises many questions on the exact details of the HOW of these various mechanisms. In this manuscript we strategized to report the observations, which, as outlined above revealed many steps and processes previously not recognized. These are the basis for our future work, but we consider the collection of novel insights reported in this manuscript as a major advance in the field.

Reviewer #3:

Specific concerns below:

1. The authors are over-reliant on the use of spectral counts for quantitation. In particular, identifying candidates in Fig. 1A based on NormSpC and not more rigorous statistical analyses of enrichment is concerning. Considering the authors use SAINT later in the manuscript, it is not clear why they did not use it in Fig 1 also instead of arbitrarily focusing on top 25 hits.

The point we were trying to make is what guided our reciprocal choice of candidates BCC1 and BCC2 for a second round of BioID baits. We would like to note that all assignments further in the paper were based on the comprehensive analysis of all data 6 BC baits, which message might have been distorted by showing the limited and, as pointed out, incomplete analysis in Fig 1. In response to this comment, we have removed this initial analysis from the manuscript and eliminated the heatmap from Fig 1.

2. It should be noted that although the use of delta spectral counts as a metric for differential abundance is crude at best and better done using more robust statistical frameworks (like SAINT or Compass), the biochemical validation of the results in this study are convincing which greatly lessens concern over the analysis of the proteomic data.

See comment above: we removed the analysis based solely on normalized spectral counts as was shown in the original Fig 1. SAINTexpress analysis was the basis for all assignments in the paper. Furthermore, as the reviewer correctly points out, were extensively experimentally validated. See reply to Reviewer 2 comment 1 for additional insights.

REVIEWERS' COMMENTS

Reviewer #1 (Remarks to the Author):

The revised submission from Engelberg and colleagues is a detailed evaluation of the basal complex and associated structures in *Toxoplasma gondii*. The study is well done and the data are convincing. The authors have addressed my previous concerns.

MINOR COMMENTS

1. Fig 3 -- the arrows and arrowheads in 3i are referred to as "red" and "blue". I think that they seem more "teal" and "orange". I suggest either changing the color names or changing the actual colors. This is confusing for the reader in an already very complicated figure.
2. Figure 5 - "Bleu" should be "Blue"
3. Figure 6d - "Tirl" should be "Tir1", also in 6g legend. And 7c.
4. Fig 9 - the chemical inhibition of microtubules with oryzalin is shown but this is not discussed in the text at all.

Point by point rebuttal of Engelberg et al, NCOMMS-21-50887B

REVIEWERS' COMMENTS

Reviewer #1 (Remarks to the Author):

The revised submission from Engelberg and colleagues is a detailed evaluation of the basal complex and associated structures in *Toxoplasma gondii*. The study is well done and the data are convincing. The authors have addressed my previous concerns.

MINOR COMMENTS

1. Fig 3 -- the arrows and arrowheads in 3i are referred to as "red" and "blue". I think that they seem more "teal" and "orange". I suggest either changing the color names or changing the actual colors. This is confusing for the reader in an already very complicated figure.

We have changed the color names as suggested by the reviewer

2. Figure 5 - "Bleu" should be "Blue"

Changed as suggested

3. Figure 6d - "Tirl" should be "Tir1", also in 6g legend. And 7c.

Changed as suggested

4. Fig 9 - the chemical inhibition of microtubules with oryzalin is shown but this is not discussed in the text at all.

Thanks for catching this, we have removed the schematic of the oryzalin treatment from Fig. 9 since, as commented by the reviewer, it is not discussed in this manuscript.